

# Ecosystem carbon transit versus turnover times in response to climate warming and rising atmospheric CO₂ concentration

Xingjie Lu[1,2,3], Ying-Ping Wang[3], Yiqi Luo[2,4] and Lifen Jiang[2]

[1]School of Atmospheric Sciences, Sun Yat-sen University, Guangzhou 510275, China
[2]Center for Ecosystem Science and Society, Department of Biological Sciences, Northern Arizona University, Flagstaff 86011, USA
[3]CSIRO Oceans and Atmosphere, Aspendale 3195, Australia
[4]Department for Earth System Science, Tsinghua University, Beijing 100084, China

*Correspondence to*: Xingjie Lu (xingjie.lu@nau.edu)



**Abstract.** Ecosystem carbon (C) transit time is a critical diagnostic parameter to characterize land C sequestration. This parameter has different variants in literatures, including a commonly used turnover time. However, neither of them has been carefully examined under transient C dynamics in response to climate change. In this study, we estimated both C turnover time as defined by the conventional stock-over-flux (i.e., Olson method) and mean C transit time as defined by the mean age of C mass leaving the system (i.e., Rasmussen method). We incorporated them into Community Atmosphere-Biosphere-Land

Exchange model (CABLE) to estimate C turnover time and transit time, respectively, in response to climate warming and rising atmospheric [$CO_2$]. Modeling analysis showed that both C turnover time and transit time increased with climate warming but decreased with rising atmospheric [$CO_2$]. The increase of C turnover time with warming was estimated to be 2.4 years with Olson method whereas the transit time increased by 11.8 years with Rasmussen method. The decrease with rising atmospheric [$CO_2$] was estimated to be 3.8 years with Olson method and 5.5 years with Rasmussen method. Our analysis based on

Rasmussen method showed that 65% of the increase in global mean C transit time with climate warming results from the depletion of fast-turnover C pool. The remaining 35% increase results from accompanied changes in compartment C age structures. Similarly, the decrease in mean C transit time with rising atmospheric [$CO_2$] results approximately equally from replenishment of C into fast-turnover C pool and subsequent decrease in compartment C age structure. Greatly different from the Rasmussen method, the Olsen method, which does not account for changes in either C age structure or composition of

respired C, underestimated impacts of either warming or rising atmospheric [$CO_2$] on C diagnostic time and potentially lead to biases in estimating land C sequestration.



## 1 Introduction

Terrestrial ecosystem plays an important role in mitigation of climate change through sequestering carbon (C) from the atmosphere. Terrestrial C storage is co-determined by C input and C transit time, which is defined as the mean age of C mass leaving the system (Luo et al., 2001; Taylor and Lloyd, 1992; Nir and Lewis, 1975; Sierra et al., 2016; Manzoni et al., 2009; Eriksson, 1971; Bolin and Rodhe, 1973). As transit time cannot be easily estimated from observation, its variant, C turnover time, has been commonly used in the literature (Sierra et al., 2016). Recent model inter-comparison study indicated that a

major cause of uncertainty in predicting future terrestrial C sequestration is the variation in C turnover time among the models (Friend et al., 2014). Up to 40% of soil C sequestration potential can be overestimated due to underestimation of C turnover time in current CMIP5 models (He et al., 2016). The C turnover time has been mostly estimated with a conventional stock-over-flux method (Carvalhais et al., 2014; Chen et al., 2013; Yan et al., 2017), which is probably first introduced by (Olson, 1963). Hereafter we call it Olson method. The Olson method is based on a steady-state assumption. In response to climate

change, terrestrial ecosystem C dynamics move away from steady states to be at dynamic disequilibrium (Luo and Weng, 2011). Estimation of turnover time with Olson method likely deviates from transit time in response to climate change (Sierra et al., 2016). It is not clear how much estimates of C turnover time deviate from mean C transit time and what cause their deviation under climate change.

The C transit time as the mean age of C mass leaving the system can be estimated only from age structure of C atoms in a multi-compartment ecosystem. In contrast, the C turnover time is estimated without any information of age structure of C atoms. Thus, C turnover time is equivalent to C mean transit time only in the autonomous (i.e., time-invariant) linear system at steady states (Sierra et al., 2016) with three conditions to be satisfied. The first condition is that C fluxes and turnover rates of each pool do not change with time (i.e., time invariant or autonomous). The second is that C turnover rate of each pool is

not a function of pool size as in linear C transfer models. The third is that C influx to each pool equals to C efflux from the pool (i.e., at steady state). However, the autonomous, linear system at steady state is a very strict condition for real-world ecosystems. Ecosystem C input via photosynthesis has diurnal variation, seasonal cycle, and inter-annual variability. C turnover time also exhibits strong seasonal variation (Luo et al., 2017). With seasonal cycles and inter-annual variability in both C input and turnover time, ecosystem C cycle is rarely at steady state. Therefore, C turnover time hardly equals C transit

time in the real world, especially when land C cycle is under transient dynamics in response to climate change.

The estimates of C transit time requires information of C age structure in ecosystems so that the mean age of the C atoms at the time they leave the system can be calculated (Manzoni et al., 2009). In a multi-compartment ecosystem, the C age within each compartment is represented by a single compartment C mean age and different compartments have different C mean ages

(Rasmussen et al., 2016). Thus, the C transit time is the weighed mean of ages of C atoms leaving different compartments according to the contributing fraction of C loss from each pool to the total C loss. This calculation of transit time hereafter is





called Rasmussen method. In response to rising atmospheric [$CO_2$], increased C input with young age into an ecosystem is usually allocated more to fast than slow turnover pools, leading to the change in C age structure in the ecosystem. The fast turnover pools usually contribute more than the slow pools to the respiratory loss. Thus, it is expected that rising atmospheric

[$CO_2$] decreases C transit time due to both changes in C age structure in the ecosystem and contributing fractions of different pools to total C loss. Although C turnover time may change in response to rising atmospheric [$CO_2$] due to changes in both C fluxes and pools, changes in C age structure are not accounted for in Olsen method. Consequently, estimates of C turnover time are likely to deviate from transit time estimates under climate change.

In this study, we aim to answer following questions: 1) How do both C turnover time and C transit times change in response to climate warming and rising atmospheric [$CO_2$]? 2) How much does the C turnover time estimated with Olson method deviate from C transit time estimated with Rasmussen method under future climate change? 3) What mechanisms cause the deviation between the two methods? 4) Which regions show the greatest biases under different climate change scenarios? To answer those questions, we incorporated both Olsen and Rasmussen methods into Community Atmosphere-Biosphere-Land surface

Exchange (CABLE) model (Wang et al., 2010; Wang et al., 2011) to estimate changes in C turnover time and transit time. We ran the modified CABLE under three climate change scenarios, climate warming only, rising atmospheric [CO2] only, and both climate warming and rising atmospheric [CO2] to compare changes in C transit time with those in C turnover time.

## 2. Materials and Methods

### 2.1 The CABLE model

CABLE is a global land surface model as described by (Kowalczyk et al., 2006) and incorporates global carbon, nitrogen and phosphorus cycles (Wang et al., 2010; Wang et al., 2011). For the sake of simplicity, this study did not activate phosphorus cycle in the model. Leaf photosynthesis, stomatal conductance, and heat and water transfer in CABLE are calculated using the two-leaf approach (Wang and Leuning, 1998).

Gross primary production (GPP) is calculated according to Farquhar photosynthesis model (Farquhar et al., 1980). Farquhar model is a biochemical model and was modified in CABLE to calculate $CO_2$ assimilation rate at canopy level as a minimum of three potential limitation processes of photosynthesis: light, enzyme and C sink. Generally, all these three photosynthetic limitations are positively related to maximal carboxylation rate ($V_{cmax}$) or maximal potential electron transport rate ($J_{max}$) and intercellular $CO_2$ concentration ($C_i$). Both $V_{cmax}$ and $J_{max}$ are temperature dependent (Leuning, 2002), which are maximized at

around 30ºC. Thus in response to warming, model usually predicts a positive response in GPP in cold and temperate regions but a negative response in GPP in hot regions. $C_i$ depends on the stomata conductance and atmospheric [$CO_2$]. GPP in CABLE positively responds to rising atmospheric [$CO_2$]. CABLE photosynthesis is also controlled by soil moisture.




Autotrophic respiration ($R_a$) in CABLE is also temperature dependent, which follows modified Arrhenius formula (Ryan, 1991; Sitch et al., 2003). In canopy scale, $R_a$ is proportional to vegetation nitrogen content and a temperature related coefficient. $R_a$

will positively respond to warming climate. Heterotrophic respiration ($R_H$) is proportional to litter and soil decomposition rate and C pool sizes. The decomposition rates in the model are controlled by soil temperature and water. The temperature response is based on a $Q_{10}$ Eqn. Decomposition rates will positively respond to warming. The water response function is from the daily time step ecosystem model (DAYCENT) (Kelly et al., 2000) and the decomposition rate positively responds to wetter soil condition.


CABLE model has three vegetation compartments (leaf, wood and root), three litter compartments (metabolic litter, structure litter and coarse wood debris), and three soil compartments (fast soil pool, slow soil pool and passive soil pool) (Wang et al., 2010).

## 2.2 Simulation design

We use the meteorological data sets from National Centers for Environmental Prediction and Climatic Research Unit – (CRU-NCEP) to drive our model. The meteorological inputs from 1901 to 2100 include temperature, specific humidity, air pressure, downward solar radiation, downward long-wave radiation, rainfall, snowfall, and wind speed. The meteorological variables of CRU-NCEP data from 1901 to 2005 were interpolated from the 6-hourly into hourly (Qian et al., 2006) and re-gridded from $0.5^o$ by $0.5^o$ to $1.875^o$ by $2.5^o$ spatial resolution. From 2006 to 2100, the hourly meteorological variables were generated from

Community Earth System Model version 1.0 (CESM) (Li et al., 2016; Hurrell et al., 2013) for Representative Concentration Pathway (RCP) 8.5.

C storage for all three scenarios (climate warming, rising atmospheric [$CO_2$] and both together) are initialized at pre-industrial steady states, which is achieved by a spin-up approach. The spin-up method cycles 10-year CRU-NCEP data (1901-1910) to

drive CABLE model, with [$CO_2$] being constant at 1901 level. A semi-analytic solution was used to accelerate spin up simulation (Xia et al., 2012).

The description of three scenarios in this study are summarized in Table 1. Simulation one (S1) fixes the atmospheric [$CO_2$] but uses changing climate forcing. Simulation two (S2) fixes climate forcing but increases atmospheric [$CO_2$]. Simulation

three (S3) uses both changing climate forcing and increasing atmospheric [$CO_2$].

## 2.3 Calculation of ecosystem C mean age

C mean age is defined as the mean time elapsed since the C atoms (current in the system) enter the system, which is important for understanding C transit time described below. Following Rasmussen et al. (2016), C mean age ($\bar{a}$) can be formulated:



$$\bar{a}(t) = \frac{\sum_{i=1}^{d} a_i(t)x_i(t)}{\sum_{i=1}^{d} x_i(t)} \tag{1}$$

In Eqn (1), $a_i$ represents the mean age of C in the $i^{th}$ compartment; $x_i$ represents C pool size of the $i^{th}$ compartment, and $d$ is total number of C compartments.

Mixing fresh C input into ecosystem old C may reduce the ecosystem C mean age. Meanwhile, C remaining in the system will age with the time. As shown by Rasmussen et al. (2016), dynamics of compartment C mean age can be described by the

following differential equation:

$$\frac{da_i(t)}{dt} = 1 + \frac{\sum_{j=1}^{d}\left(a_j(t)-a_i(t)\right)b_{ij}k_j(t)x_j(t)-a_i(t)s_i(t)}{x_i(t)} \tag{2}$$

In Eqn (2), $s_i(t)$ is direct C input rate from net primary production to the $i^{th}$ compartment in g C m$^{-2}$ year$^{-1}$, $b_{ij}$ is the proportion of decomposed carbon from $j^{th}$ compartment to be transferred to the $i^{th}$ compartment. $k_j$ is the decomposition rate of the $j^{th}$ compartment, the unit is year$^{-1}$. Thus, change in compartment C age depends C aging, network C transfers among pools with

different ages, and C input.

With a time step $\Delta t$, the C transferred from the $j^{th}$ compartment to the $i^{th}$ compartment ($F_{ij}$) equals to $b_{ij}k_j(t)x_j(t)\Delta t$ and C input ($S_i$) equals to $S_i(t)=s_i(t)\Delta t$, Eqn (2) can be rewritten in a finite element form to represent C age dynamics:

$$\Delta a_i(t) = \Delta t + \frac{\sum_{j=1}^{d}\left(a_j(t)-a_i(t)\right)F_{ij}-a_i(t)S_i(t)}{x_i(t)} \tag{3}$$

In Eqn (3), the first term, $\Delta t$, indicates natural C aging. the second term, $\frac{\sum_{j=1}^{d}\left(a_j(t)-a_i(t)\right)F_{ij}-a_i(t)S_i(t)}{x_i(t)}$, represents the mean age change of the $i^{th}$ compartment due to mixing with transferred C from other compartments or external C input (i.e., NPP).

After the C cycle spin up, we ran age-structured CABLE to obtain initial steady state of each compartment C age by cycling 10-year (1901-1910) C fluxes and pool sizes from CABLE to force Eqn (3) until the changes of C compartment mean age are

less than 0.1% between two successive cycles.

**2.4 Rasmussen ecosystem C transit time**

C transit time is defined as the average time for a C atom spend in the ecosystem until its exit, or the time from entering the ecosystem to leaving the ecosystem (or residence time, (Luo et al., 2001)). For a multiple-compartment system, the mean C transit time, $\bar{\tau}_R$, estimated by Rasmussen method can be calculated as following (Rasmussen et al., 2016):

$$\bar{\tau}_R(t) = \frac{\sum_{i=1}^{d} a_i(t)x_i(t)(\sum_{j=1}^{d} b_{ji})k_i(t)}{\sum_{i=1}^{d} x_i(t)(\sum_{j=1}^{d} b_{ji})k_i(t)} \tag{4}$$





when $i = j$, $b_{ii} = -1$, indicating one unit of C exited from the $i^{\text{th}}$ compartment. When $i \neq j$, $b_{ji}$ represents the proportion of exited C of the $i^{\text{th}}$ compartment transferred to $j^{\text{th}}$ compartment. $\sum_{j=1}^{d} b_{ji} = 0$ when the exited C from the $i^{\text{th}}$ compartment is fully transferred to all the other compartments, such as litterfall from plant to litter compartments, without C loss. $\sum_{j=1}^{d} b_{ji} < 0$ when the exited C from the $i^{\text{th}}$ compartment is partly transferred to the other compartments, such as litter or soil C

decomposition, with the rest lost to the atmosphere via respiration. The denominator is the total amount of C loss from the ecosystem. The numerator is the sum of respired age-mass C.

**2.5 Components of Rasmussen C transit time and their changes**

Equation (4) can be re-organized as:

$$\bar{\tau}_R(t) = \sum_{i=1}^{d} a_i(t) f_{\text{hr},i}(t) \tag{5}$$

when we define fraction of the total C loss from the $i^{\text{th}}$ compartment ($f_{\text{hr},i}$) as:

$$f_{hr,i}(t) = \frac{x_i(t)(\sum_{j=1}^{d} b_{ji}) k_i(t)}{\sum_{i=1}^{d} x_i(t)(\sum_{j=1}^{d} b_{ji}) k_i(t)}$$

Equation (5) indicates ecosystem C transit time has two components: compartment C age ($a_i$) and the fractional composition of respired C ($f_{\text{hr},i}$). Compartment C age as represented by Eqn (2) changes due to C mixing with those transferred with other compartments or from external input.


According to Eqn (5), the change in ecosystem C transit time $\bar{\tau}_R$ can be attributed to the change in compartment C age (change in C age structure) and the change in respired C composition as (See Supplementary Information for details):

$$\Delta\bar{\tau}_R(t) = \sum_{i=1}^{d} a_i(t)\Delta\left(f_{\text{hr},i}(t)\right) + \sum_{i=1}^{d} f_{\text{hr},i}(t)\Delta\left(a_i(t)\right) + o(a_i(t), f_{\text{hr},i}(t)) \tag{6}$$

The first term in Eqn (6) refers to C transit time change due to change in respired C composition. If the fraction of respired C

from fast-turnover pool decrease, the ecosystem mean C transit time may increase because more respired C comes from slow-turnover pools with older C ages. The second term refers to C transit time change due to change in compartment C age structure. If more young-age C influx into a compartment than the C efflux as under elevated $CO_2$, C age structure in the compartment will become younger (i.e., young-age C replenishment). Subsequently, ecosystem mean C transit time will reduce. The third term refers to residuals that cannot be explained by previous two terms.

**3. Results**

**3.1 Global steady-state patterns of ecosystem C transit time**

The global ecosystem C transit time at steady state estimated by Rasmussen method generally shows a latitudinal variation pattern (Fig. 1). The high values (greater than 70 years) are simulated not only in high latitude regions, such as northern Russia, northern Europe, and northern Canada but also in high altitude regions such as Tibet plateau. Small values in C transit time



(less than 30 years) are simulated in tropical rainforest, such as Amazon forest, Conga forest, and Indonesia forest. Ecosystem C transit times in some grass lands in middle-south Africa, south America, Southern Great Plains of US, and central north Australia (savanna) sometime are even smaller than that in tropical forest. The spatial patterns of the ecosystem C mean age are quite similar with the patterns of C transit time. However, the magnitude is significantly higher than ecosystem C transit time. The ecosystem C mean age ranges from 118 years to 7952 years, whereas ecosystem C transit time ranges only from 13
years to 341 years.

The global latitudinal pattern of C transit time estimated from Rasmussen method in 1982-2005 is consistent with the observation-based pattern of turnover time (Fig. 2). The latter is estimated at each grid cell globally by Olson method to divide ecosystem C storage by gross primary productivity (GPP) (Carvalhais et al., 2014). The magnitude of the estimate is mostly
within the uncertainty range of the observation-based pattern. We compared estimated C transit time in 1982-2005 with the turnover time, partly to match modelled values with contemporary observations, which is  based on the fact that terrestrial C cycle is still approximately at a quasi-steady state between 1982 and 2005. Over the 1980s and 1990s, the annual average of global net land carbon sink estimated from Global Carbon Project (GCP) is about 0.8 GtC yr$^{-1}$ with an uncertainty of 0.6 GtC yr$^{-1}$. As a reference, the annual average of net land carbon sink in recent decade (2007-2016) is 2.3 GtC yr$^{-1}$ with an uncertainty
of 0.7 GtC yr$^{-1}$ (Le Quere et al., 2018). The net change of global land carbon in 1980s and 1990s is not that significant, which indicates land C cycle has not moved away too far from the steady state. Annual C turnover time using Olson method theoretically equals to C transit time from Rasmussen method when C cycle is close to the steady state (Sierra et al., 2016).

**3.2 Responses of global C mean transit time to climate change**

In 200-year simulation, global ecosystem C mean transit time increased by 11.8 years in response to climate warming (S1) and
decrease by 5.6 years in response to rising atmospheric [$CO_2$] (S2) (Fig. 3a). When climate warming and rising atmospheric [$CO_2$] forced together (S3), C transit time decreased by 1.6 years. The increase in C transit time in S1 is not significant in the 20$^{th}$ century but substantial in the 21$^{st}$ century. Oppositely, the decrease in C transit time in S2 is steady before 2060 but slow down afterward. Mean C transit time in S3 decreases but with a smaller magnitude than that for S2 in the 21$^{st}$ century.

Across all the three scenarios, the most majority (over 93.4%) of the change in C transit time can be explained by the combined changes in compartment C age structure and respired C composition. Changes in the compartment C age structure and the respired C composition both significantly contributed to the total change in global C transit time. However, the contribution fraction varied among the three scenarios at different time. In climate warming scenario (S1), respired C composition changes contribute about 70% of the increase in C transit time in the 21$^{st}$ century (Fig. 3b). In the rising atmospheric [$CO_2$] scenario
(S2), respired C composition change and C age structure change contribute equally (Fig. 3c). When coupling climate warming and rising atmospheric [$CO_2$] together in S3, respired C composition change significantly contributes only in the middle of



200-year simulation (around year 2000), but little at the end of the 21$^{st}$ century. The contribution of C age structure change to the change in C transit time gradually increases.

The increase in C transit time in climate warming scenario (S1) is the most significant from low latitude regions in south America and Africa (Fig. 4a). Respired C composition change explains most of these regional changes (Fig. 4c). The decrease in C transit time in rising atmospheric [CO2] scenario (S2) is evenly simulated all over the world (Fig. 4d). Respired C composition change also plays an important role in most regions except for north Africa with little vegetation coverage. The C transit time in combined climate warming and rising atmospheric [$CO_2$] scenario (S3) mostly decrease in northern

hemisphere, but increase in some tropical grassland regions in South America and Africa (Fig. 4g). In those regions where C transit time decrease, compartment C age structure change due to fresh C replenishment explain most of the change in C transit time.

### 3.3 Global C turnover time and its bias

Similar to the changes in C transit time estimated by Rasmussen method, the global C turnover time estimated by Olson method

increases with climate warming and decreases with rising atmospheric [$CO_2$] (Fig. 5a). However, the magnitude substantially differs between these two methods (Fig. 3a, 5a). In response to climate warming (S1), global ecosystem C turnover time increases by only 2.4 year at end of the simulation, which is only one-fifth of the increase in C transit time (11.8 year). In response to rising atmospheric [$CO_2$] (S2), global C turnover time decreases by 3.7 year, whereas C transit time decreases by 5.6 year. In response to the coupled scenario (S3) where climate warming and rising atmospheric [$CO_2$] force together, global

ecosystem C turnover time decreases by 4.5 year, while C transit time decreases by only 1.6 year.

In 1901, the global C turnover time is about 0.5 year longer than the C transit time (Fig. 3a, Fig. 5a). Theoretically, C turnover time equals transit time when land C cycle is at steady state. The offset at the initial state of simulations probably results from C seasonal cycles, which is not at steady state. The underestimates of the change in C turnover time relative to C transit time

increases in climate warming scenario (S1) by up to 9.4 years in the end of the 21$^{st}$ century, which is 79.6% of the total increase in C transit time (Fig. 5b). In rising atmospheric [$CO_2$] scenario (S2), the bias constantly grows to about 1.9 years, a 27.7% of the underestimated decrease in C turnover time. In climate warming and rising atmospheric [$CO_2$] scenario (S3), the change in C turnover time is overestimated by 2.9 years or 181.1% in relative to the change in C transit time in 2100 (Fig. 5b, 5c).

### 3.4 Latitudinal variation in C turnover time and its bias

Latitudinal patterns in C transit time and C turnover time at the initial state in 1900 are nearly the same. Steady state estimates are both from 20 years in low latitude to 100 years in high latitude (Fig. 6a, 6d). However, significant bias still exists in high latitudes (north of 60ºN and south of 50ºS) (Fig. 6g). The underestimates of C turnover time can be up to 10 years in permafrost regions, which is about 8% of C transit time. In other area, bias of turnover time is less than 0.5 years.





Changes in C turnover time and C transit time deviate in different regions in response to climate warming (S1) (Fig. 6b, 6c, 6e, 6f). In temperate and tropical regions, C transit time significantly increases, while C turnover time also increases but in a much smaller magnitude. In tropics, C transit time increases by 13 years in 2100, up to 60% of the initial value in 1900, whereas C turnover time increase by only 2 years. In the permafrost region, C transit time slightly decreases (Fig. 6b and c) but C turnover time significantly decreases by several decades in the high latitude (Fig. 6f). In some regions between 40°N and

60°N, C transit time increases but turnover time decreases in response to climate warming. C turnover time overall changes less than C transit time in the S1 scenario. Warming-induced changes in C turnover time is underestimated by 5% at the high latitude of the southern hemisphere to 50% at the low latitude (Fig. 6h), which range from 2 to 29 years (Fig. 6i).

In response to rising atmospheric [$CO_2$] (S2), both C turnover time and transit time decrease. The magnitude of changes for

both of them are generally greater at the mid latitudes than those at either low or high latitudes (Fig. 6b, 6e). At most latitudes, C turnover time decrease less than C transit time, leading to the positive bias (Fig. 6h and 6i). The deviation of the change is higher in the low than high latitude. In response to rising atmospheric [$CO_2$], the underestimate of the decrease in C turnover time is by at most 2 years in absolute bias or 10% in relative bias (Fig. 6h, 6i).

In climate warming and rising atmospheric [$CO_2$] scenario (S3), C turnover time and C transit time decrease at most of the latitudinal regions except for some tropic areas (Fig. 6b, 6c, 6e, 6f). The decrease in C turnover time is more than that in C transit time (Fig. 6h, 6i). Especially in high latitudes where covered by permafrost soil, the difference in changes is much more significant. C turnover time is reduced by up to three decades (Fig. 6f) or 35% (Fig. 6e), whereas C transit time shows nearly no relative changes in those. Bias in these areas can be up to 27 years (Fig. 6i).

**4. Discussion**

**4.1 C transit time and its two components**

Changes in C transit time can be explained by its two components: the respired C composition and compartment C age structure. The first component is to account for different contributions of respired C from different pools to total ecosystem C loss. Previous studies have demonstrated that pathways of respiring C from multiple compartments vary with global change factor

(Luo et al., 2001). Results from this study provide more spatial details about where C transit time change due to respired C composition change. For example, over 80% of the increase in C transit time under warming is explained by respired C composition change in the South America grassland region (Fig. 4a). In contrast, change in respired C composition only accounts for approximately 10% of the increase in C transit time under warming in the boreal and permafrost region of North America.




The second component is the C age structure, primarily from change in C mean age of individual pool modified by relative fraction of each pool. In coupled climate warming with rising atmospheric [$CO_2$] scenario (S3), C age structure change primarily contributes to the C transit time response in most global regions in 2100 (Fig. 4h). In this scenario, ecosystem mean C transit time decreases by 1.6 years. The decrease in C transit time results from increased young-age C uptake with rising

atmospheric [$CO_2$], which is more than the increased young-age C loss with warming. A previous study has also shown that models with multiple pools usually have a more heterogeneous C age structure and thus can store extremely older C than a single pool model (Manzoni et al., 2009).

## 4.2 Bias arising from estimated C turnover time

C turnover time estimated by stock-over-flux (i.e., Olson method) has been widely used to quantify ecosystem C cycle partly

because both ecosystem C storage and C flux can be easily measured (Sanderman et al., 2003; Chen et al., 2013; Carvalhais et al., 2014; McCulley et al., 2004; Raich and Schlesinger, 1992; Yan et al., 2017). The C turnover time estimated by Olson method has been theoretically shown to equal C transit time estimated by Rasmussen method at steady state but they deviate under non-steady states (Sierra et al., 2016). This study illustrates how much deviation occurs between C transit time and C turnover time in response to three scenarios of climate change. Our results show that even at initial steady state, global

ecosystem C turnover time is slightly greater than C transit time by 3%. This is because the steady state reached by spin-up does not mean the terrestrial C cycle system is completely at equilibrium. Seasonal variations of ecosystem C uptake and turnover still lead to periodical oscillation of the terrestrial C cycle.

In transient state, the changes in C transit time and C turnover time differ the most in climate warming scenario (S1). Tropical

and permafrost regions contribute the most of the deviation (Fig. 6h, 6i). In tropical and subtropical regions, C transit time increases by about 60% (Fig. 6b) while C turnover time increases by 20% or less (Fig. 6e). The great difference between changes in C transit time and turnover time is due to their different assumptions. In response to climate warming, composition change in respired C contributes most to the change in C transit time in tropical regions. However, Olson method assumes the whole ecosystem C as one homogenous pool, even if both plant and soil C can be extremely heterogeneous. This homogeneity

assumption ignores the composition changes in respired C, which causes up to 80% of change in C transit time.

In permafrost regions, C transit time slightly decreases by up to 10%, whereas C turnover time considerably decreases by over 30% in response to climate warming. Warming significantly increases soil respiration due to permafrost thaw, whereas the change in permafrost ecosystem C pool size is relatively small. Thus, C turnover time estimated by Olson method significantly

decreases. C transit time slowly responds to climate warming because the young-age C input added to permafrost ecosystem is relatively small compared to large C storage in this area and C age structure does not change much. These big deviations between C turnover time and C transit time in tropical and permafrost regions suggest that future C cycle analysis based on turnover time likely leads to strong biases as it does not represent transient C dynamics in multi-pool ecosystems.



### 4.3 C transit time versus turnover time under other global change scenarios

This study has illustrated how C transit time and turnover time deviate under climate warming and rising atmospheric [$CO_2$]
scenarios. Those deviations may become even bigger under other global change scenarios. For example, land use change and
fire can drive ecosystems out of steady state to be at disequilibrium (Luo and Weng, 2011). Clearcut of forest or forest fire
removes at least the aboveground wood C pools and thus greatly changes both the total C stock and NPP, leading to a large
change in C turnover time (Wang et al., 1999; Zhou and Luo, 2008). Clearcut of forest or forest fire also changes age structure

and composition of respired C from different pools within the ecosystem, resulting in change in C transit time. Such a
disturbance usually drives ecosystem to a stronger degree of disequilibrium than climate change does, the deviation between
turnover time and transit time should be bigger under a severe disturbance than climate change. It is consistent with our results
that C transit time and turnover time deviates more significantly when an ecosystem is further away from equilibrium (Fig. 5).

In the real world, land C cycle is always at dynamic disequilibrium due to cyclic environmental conditions (e.g., diurnal,
seasonal, and interannual variability), directional global change (e.g., climate warming, rising atmospheric CO2 concentration,
altered precipitation, and nitrogen deposition), recursive disturbance-recovery cycles, shifted climatic and disturbance regimes,
and vegetation changes (Luo and Weng, 2011). Thus, the estimated C turnover time by Olson method is expected to differ
from the C transit time by Rasmussen method at any time point and at any spatial location. The degree of deviation between

C turnover time and transit time may vary.

In addition to various agents to cause ecosystem to be at disequilibrium, deviation between estimated C transit time and
turnover time is also dependent model structure. Vertically resolved soil C models, for example, includes vertical C mixing
and depth-dependent C decomposition rates (Koven et al., 2013; Huang et al., 2018). Representation of vertically resolved

processes likely increase soil heterogeneity. When warming induces deep soil thaw and increases deep soil decomposition, the
fraction of respired C from deep layer with old-age C increases. The C transit time estimated by Rasmussen method together
with a vertically resolved model may substantially increase whereas C turnover time, which implicitly assumes ecosystem as
one homogeneous pool, may not respond much.

### 4.4 Estimation of C transit time in the real world

Previous studies have argued that C transit time is conceptually sounder than C turnover time (Rasmussen et al., 2016; Sierra
et al., 2016). In this study, we have shown that the C turnover time can substantially deviate from the transit time in response
to climate change and other environmental change. However, C turnover time can be easily calculated from C stock over flux,
both of which can be easily measured. In contrast, C transit time cannot be easily estimated from field measurements. Equation
(5) indicates that we need data from measurement of C mean ages ($a_i$) and fractional composition of respired C ($f_{hr,i}$) in

individual C pools in order to calculate ecosystem mean C transit time ($\bar{\tau}_R$). Neither $a_i$ nor $f_{hr,i}$ can be easily measured in field.

Thus, our research community faces a tremendous challenge to estimate a conceptually sound and scientifically important parameter.

In the past, radiocarbon $^{14}$C has been used to quantify C mean ages of various litter and soil pools (Gaudinski et al., 2000).
Measured soil respiration in response to elevated $CO_2$ treatment in Duke Forest has been decomposed to various fractional composition using a deconvolution method or inverse analysis (Luo et al., 2001). It appears that estimation of C transit time in the real-world ecosystems requires measurement of isotope signatures in different litter and soil fractions together with measurement of respiration from soil surface and soil components. Those measurements, together with many other data sets, may need to be analyzed to estimate C mean ages, fractional composition of respired C in individual C pools, and then
ecosystem mean C transit time ($\bar{\tau}_R$) using some innovative ways, such as data assimilation.

## 5. Conclusions

This study explores how global ecosystem C transit time deviate with the turnover time under climate warming and rising atmospheric [$CO_2$]. Although both global ecosystem C transit time and turnover time increase in response to climate warming and decrease in response to rising atmospheric [$CO_2$], the deviations increase with time in all scenarios. In 2100, the deviations
are high in tropical regions under climate warming scenario (S1) and rising atmospheric [CO2] scenario (S2), and in permafrost regions under S1 and combined change scenario (S3).

The changes in C transit time results from both the C age structure changes and composition changes in respired C in multi-pool ecosystems. The C age structure changes mainly depend on young-age C replenishment from external C input. The
composition change is due to differential responses of various C pools to climate warming and rising atmospheric [$CO_2$]. However, C turnover time assumes ecosystem as one homogeneous pool, and it does not account for changes in age structure and contribution fractions of different pools to ecosystem respiration. Thus, C transit time is a better parameter than C turnover time to characterize C cycle in multi-pool ecosystems, especially when they are at transient states.

However, C transit time cannot be easily measured because it requires information of the C age structure and composition of respired C. Both of them are usually not measurable in field studies. Radiocarbon $^{14}$C measurement in the field has the potential to offer information on mean C ages in various pools. It is not easy, either, to estimate contribution fractions of different pools from measured ecosystem or soil respiration to respired C. We may have to combine compartment models with different types of measurements via data assimilation techniques to estimate both age structure and composition of respired C before we can
estimate ecosystem C transit time.

**Acknowledgments**





This research was financially supported by the post-doctoral fellowship from the CSIRO Office of Chief Executive to X.J.L
and U.S. Department of Energy grants DE-SC0008270, DE-SC0014085, and U.S. National Science Foundation (NSF) grants
EF-1807529 and OIA-1301789 to Y.Q.L EcoLab.

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

**Table 1** Summary of scenarios and forcing data.

| Scenario name | Simulation abbreviation | Climate forcing | CO$_2$ data |
|---|---|---|---|
| **Climate warming scenario** | S1 | Climate warming[#] | Pre-industrial[**] |
| **CO$_2$ direct effect scenario** | S2 | Pre-industrial[##] | CO$_2$ increase[*] |
| **Full effect scenario** | S3 | Climate warming[#] | CO$_2$ increase[*] |



# Climate warming forcing data from 1901 to 2005 uses CRU-NCEP dataset. The forcing data from 2006 to 2100 uses CESM output under Representative Concentration Pathways with radiative forcing increased by 8.5 W m$^{-2}$ (RCP8.5).

## Pre-industrial climate forcing repeatedly uses one-year climatology data averaged over 1901 to 1910 from CRU-NCEP dataset.

* $CO_2$ concentration data are from 200-year CMIP5 dataset under historical and future scenario (RCP8.5).

** Pre-industrial $CO_2$ concentration is from CMIP5 dataset for the year 1901.



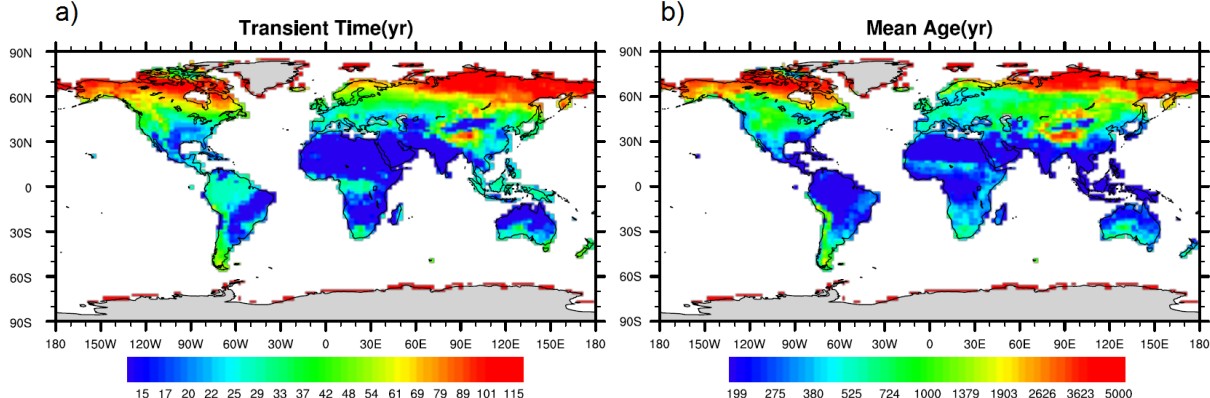

**Figure 1 Global map of a) carbon transit time and b) carbon mean age are the average over 1901 to 1910 at each grid cell estimated by Rasmussen method.**






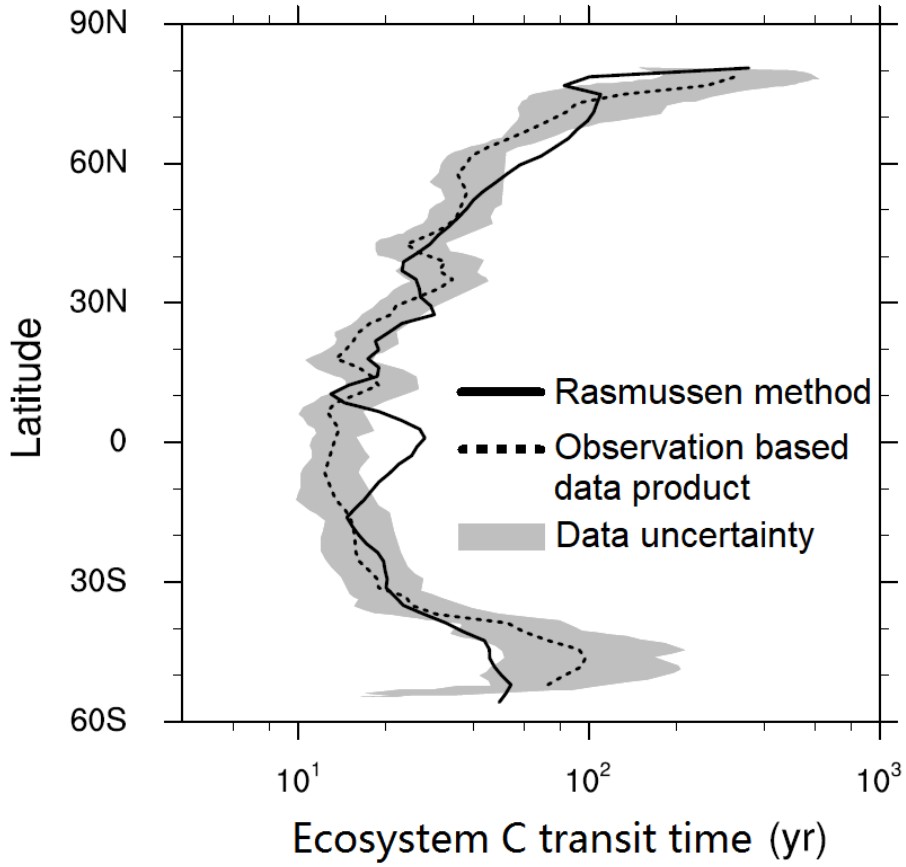

**Figure 2 Validation of simulated latitudinal variation pattern in ecosystem C transit time. Comparison of the ecosystem C mean transit time from 1982 to 2005 as estimated in this study with the estimates from observation (Carvalhais et al., 2014). Grey area indicates the uncertainty range of observation-based data.**






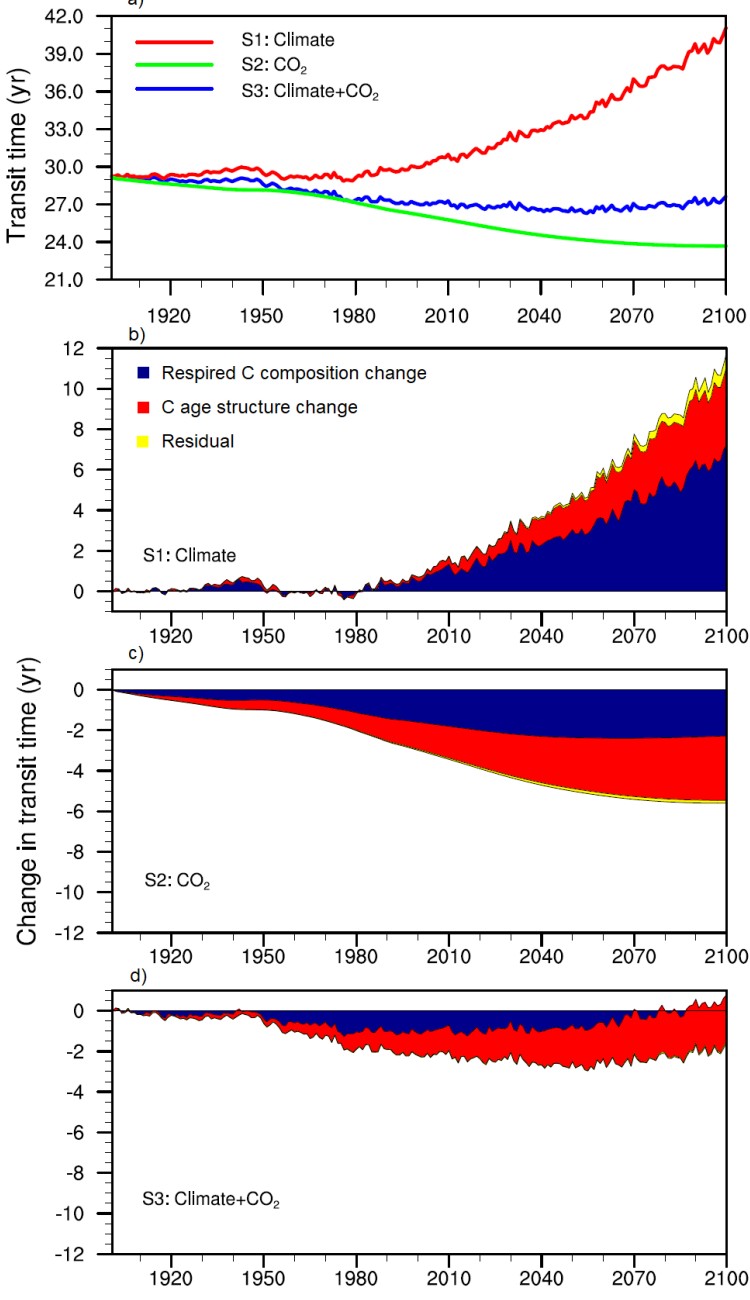

**Figure 3 CABLE simulated changes of global C transit time using Rasmussen method for each of the three scenarios in a): S1: climate warming scenario (red line); S2: rising atmospheric [CO₂] scenario (green line), and S3: Combining climate warming and rising atmospheric [CO₂] scenario (blue line). The changes in global ecosystem C transit time were separated into three contributions based on Equation (6): contribution from respired C composition change, contribution from C age structure change and residual (b-d).**






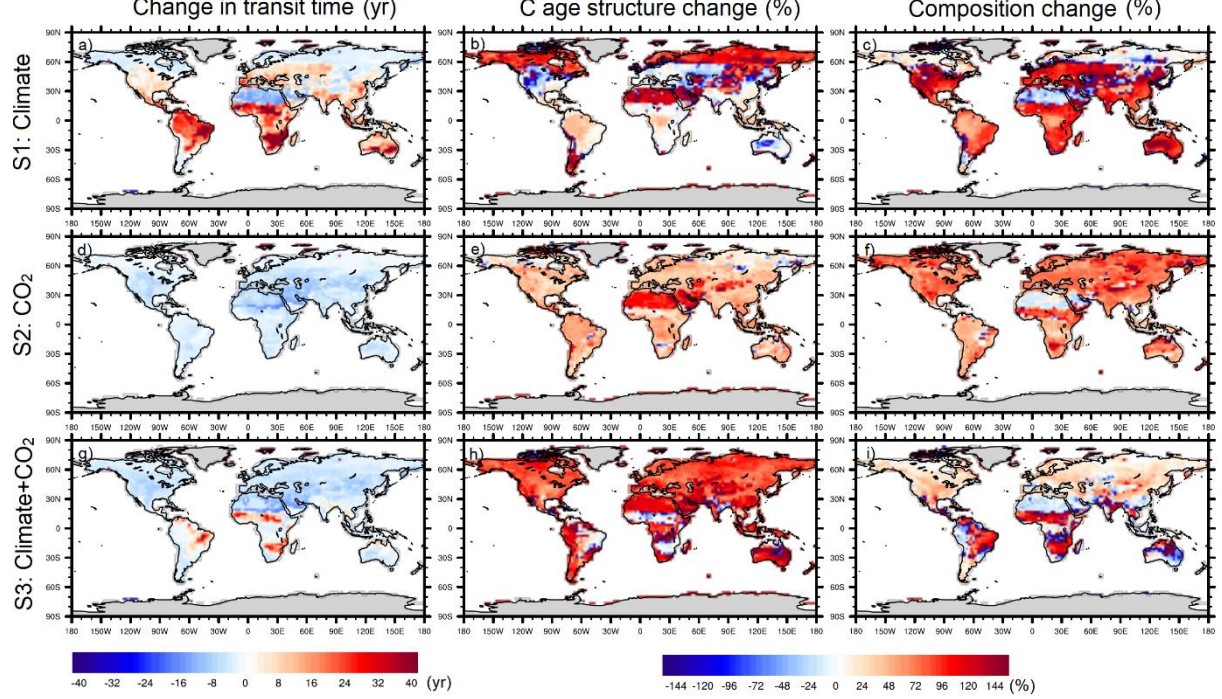

**Figure 4 Global map of the change in C transit time in three scenarios, a) S1: climate warming scenario; d) S2: rising atmospheric [CO2] scenario, and g) S3: Combination of climate warming and rising atmospheric [CO2] scenario. In these three scenarios, contribution from C age structure change and contribution from respired C composition change are also estimated in relative to the change in C transit time (S1: b) and c); S2: e) and f); S3: h) and i)). The calculation of contribution from C age structure change and contribution from respired C contribution change are based on Equation (6). The positive contribution indicates the C age structure change or composition change leads to C transit time change towards the same direction.**




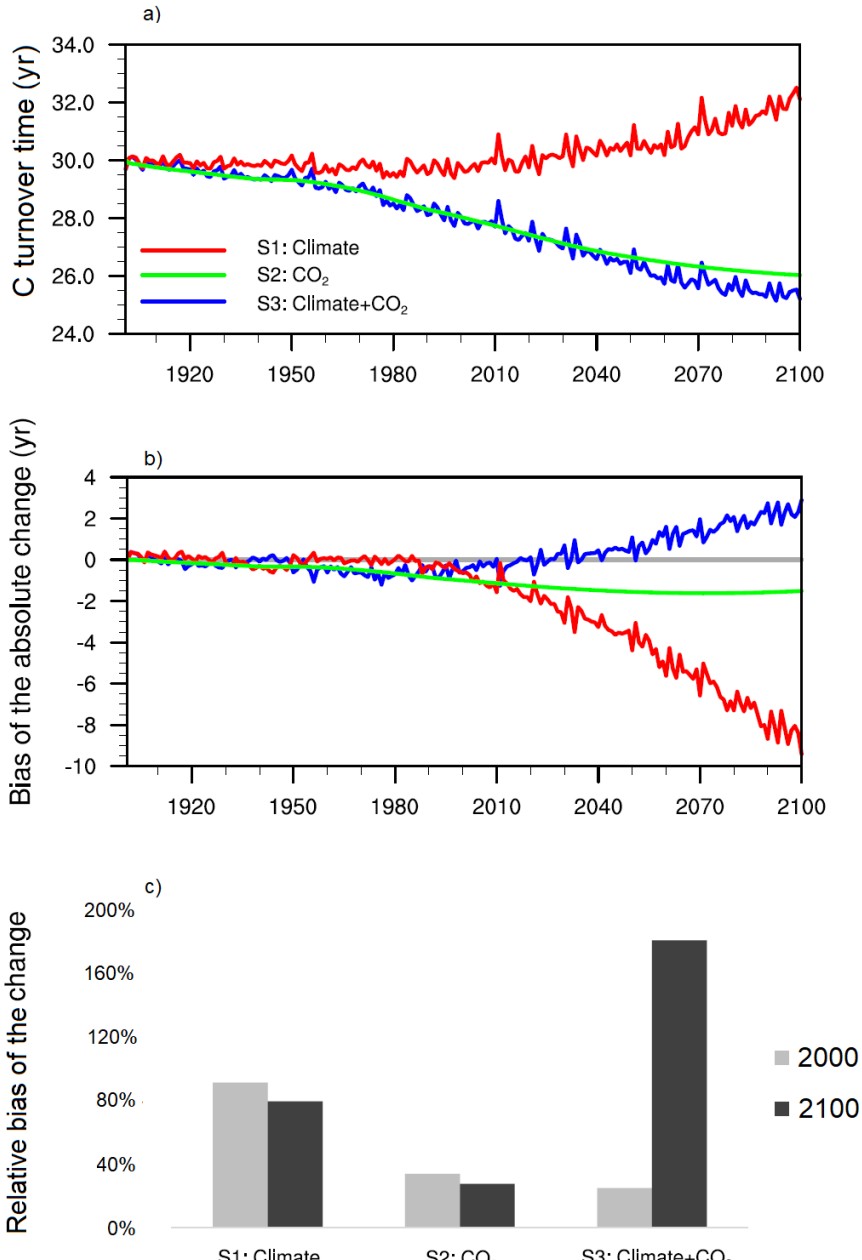


**Figure 5 a) Changes of global Olson C turnover time (stock-over-flux) in three scenarios, S1: climate warming scenario (red line); S2: rising atmospheric [CO$_2$] scenario (green line), and S3: Combination of climate warming and rising atmospheric [CO$_2$] scenario (blue line). b) The bias of the change in C turnover time ($\Delta\tau_o$) was estimated relative to the change in Rasmussen C transit time ($\Delta\tau_R$): ($|\Delta\tau_o| - |\Delta\tau_R|$). Positive indicates more change in C turnover time than C transit time. Grey line represents the reference of**
**no bias. c) The relative bias of the change in C turnover time in year 2000 and 2100 was also estimated relative to the change in C transit time: $\frac{(|\Delta\tau_o|-|\Delta\tau_R|)}{|\Delta\tau_R|} \times 100$%.**



**Figure 6 a)** Latitudinal variation in Rasmussen C transit time ($\tau_R$) at steady state and **b)-c)** its change are compared to **d)-f)** Olson C turnover time ($\tau_o$). The changes between 2090s and 1900s are estimated by **c)**, **f)** absolute value: $\Delta\tau = (\tau_{2090s} - \tau_{1900s})$ and by **b)**, **e)** relative value: $\Delta\tau_r = \frac{\Delta\tau}{\tau_{1900s}}$. **g)** The bias of C turnover time in relative to C transit time is estimated by $(\tau_o - \tau_R)$ at steady state. In relative to C transit time, the bias of the change in C turnover time are estimated by **h)** absolute bias $(|\Delta\tau_o| - |\Delta\tau_R|)$ and **i)** relative bias in $\frac{(|\Delta\tau_o| - |\Delta\tau_R|)}{|\Delta\tau_R|}$. All variables are compared in three scenarios: **S1:** only climate warming scenario (red line); **S2:** rising atmospheric [CO$_2$] scenario (green line), and **S3:** Combination of climate warming and rising atmospheric [CO$_2$] scenario (blue line). Grey lines in b), c), e) and f) represent the reference lines of no change and those in h) and i) represent reference line of no bias.
