# Peer review of "Ecosystem carbon transit versus turnover times in response to climate warming and rising atmospheric CO2 concentration"

_Biogeosciences, 2018_

## Referee Comment (RC1) · Anonymous Referee #1 · 18 May 2018

X. Lu, Y.-P. Wang, Y. Luo, and L. Jiang

"Ecosystem carbon transit versus turnover times in response to climate warming and rising atmospheric $CO_2$ concentration"

submitted to Biogeosciences, April 2018

May 18, 2018

It is well known that carbon turnover time as computed by carbon stock divided by carbon flux has a well defined meaning only for stationary states, while the meaning of transit time as mean age of carbon released from the system remains valid also for non-stationary states. On this background the submitted paper investigates how strongly transit and turnover times deviate in historical and RCP8.5 scenario simulations. The simulations are performed using the land surface model CABLE in offline simulations forced by CRU-NCEP (historical) and CLM (scenario) data. To separate the physical and biogeochemical effect of $CO_2$ on transit and turnover times, separate simulations are performed where either the temperature forcing or the photosynthethically relevant $CO_2$ level are kept fixed. To determine transit time using the approach by Rasmussen et al. (2016), the authors equipped CABLE with a diagnostic to follow changes in carbon stocks of all pools of their land carbon model and the fluxes between them. The authors show that the expression for changes in transit time based on this approach can be separated into two components, one arising from changes in the mean age of the carbon in the different pools (abbreviated in the following as MAC – Mean Age Change), the other arising from changes in the age composition of the carbon fluxes 'respired' from the different carbon pools (abbreviated in the following as ACC – Age Composition Change). The authors show how the MAC and ACC contributions to transit time change in their scenario simulations.

**Major Remarks**

**1) The study is not well motivated**
In the abstract the authors motivate their study by writing that considering transit and turnover times "neither of them has been carefully examined under transient C dynamics in response to climate change". This is not a very convincing argument for their study since (i) the study should not be published if its contents would not be new, and (ii) that something hasn't been done doesn't qualify it as scientifically relevant. And also the introduction is not clear about the motivation of the study, except that from the subtext the authors let arise the impression that results from other studies using turnover time for a non-steady state situation cannot be trusted. Here e.g. a study by He et al. (2016) is cited with the result that in CMIP5 simulations the "soil C sequestration potential can be overestimated due to underestimation of C turnover time". But in this study "turnover

times" are decay parameters of a model and not a diagnostic turnover time computed by carbon stock divided by carbon flux so that this study is not suitable for motivating the study reviewed here. A similar remark concerns the study of Friend et al. (2013) cited in the introduction: Friend et al. indeed calculate turnover time as carbon stock divided by carbon flux but as a diagnostic to test the validity of their simple carbon model, but not for drawing any other conclusions from it that could be improved by using transit time. Hence, in my opinion the authors handle the cited literature inappropriately to motivate their study and thereby give a wrong impression of their relevance.

**2) The relevance of the results of the study is unclear**
According to section 5 ('Conclusions') the study has two major results. The first (lines 357-361) concerns the development of transit and turnover time in their simulations, in particular that they increasingly deviate from one another during the 21st century, and that the deviations are stronger in some regions than in others. But what to conclude from this? Is this a useful knowledge?
The second result (lines 363-368) is that transit time can be separated into contributions from MAC and ACC because the residual is small (see eq. (6) and Fig. 3). Therefore on first sight I indeed thought that this separation is an interesting idea that could help to understand better how transit time behaves under different forcings. The authors claim (lines 364/365) that MAC is determined by carbon input and ACC by "differential responses of various C pools to climate warming and rising atmospheric [$CO_2$]". While this latter formulation is rather cryptic, I take from it that the authors think that one could pin down what affects MAC and ACC so that e.g. for scenarios of different $CO_2$ and/or temperature rise one could understand why transit time would develop differently. But I very much doubt that this separation helps understanding anything since there is no way to see how MAC and ACC are separately affected by the carbon inputs or the forcings: this is because MAC and ACC are not independent from one another since they are both derived from the same development of carbon stored in the compartments ($x_i(t)$ in eq. (2) and eq. line 166). Hence a change of the carbon input into the system changes both MAC and ACC, and also a change in pool turnover time parameters by a changing climate (temperature, moisture) changes both MAC and ACC. Only if one could understand how climate and $CO_2$ act differently on MAC and ACC this separation could contribute to a better understanding of transit time development in transient simulations. That MAC and ACC have no individual meaning can also be seen directly in the simulation results: In the simulation with both forcings combined (simulation S3) the contribution to transit time from ACC is not even approximately the sum of the ACC contributions from the simulations with forcings separated (S1, S2), and the same is true for MAC. Hence, the behaviour of simulation S3 cannot be understood as combination of results from S1 and S2. – In conclusion, I think this separation is only technical and pretty useless. In order to convince me from the opposite, the authors had to show me a case where it leads to an improved understanding.
Concernig the other remarks related to the second result in lines 366-368, I think they are all wrong: (i) The calculation of turnover time by dividing stocks by fluxes is not

assuming anything, it is simply a diagnostic that in the case of stationary states has a well defined meaning, but can, as a diagnostic, still be a useful concept (see my remark on the study by Friend et al. (2013) above). (ii) Surely turnover time changes when MAC and ACC change, so that contrary to the authors claim it accounts for such changes. Hence (iii) contrary to the claim by the authors one cannot conclude that transit time is a "better parameter". – A similar claim is found in the last scentence of the abstract where the conclusion is even weirder by saying that the use of turnover time instead of transient time may "lead to biases in estimating land C sequestration" – how could the mere calculation of a time scale affect the estimation of C sequestration?

**3) Some suggestions for improving the paper**
There is one result of the paper – surprisingly not mentioned in the conclusions – that in my opinion makes an important contribution to land carbon research: This is the comparison of the CABLE results for transit time with the observational results by Carvalhais et al. (2014) in Fig. 2. When the Carvalhais et al. paper appeared, I thought it's nice that they produced a map of stocks divided by fluxes so that this turnover time can be used as a diagnostic to easily compare with results from model simulations. But with the study under review here, we now know that despite the non-stationarity of todays carbon cycle, turnover times agree well with transit times (Fig. (5)) so that the observational turnover times of Carvalhais et al. (2014) can indeed be interpreted as proper carbon ages. And that the zonal distribution of CABLE results matches those of Carvalhais et al. quite well provides additional credit to this conclusion.
Hence, what I propose is that you focus your study on the question to what extend the observational estimates of turnover time by Carvalhais et al. (2014) (and if possible also those by Bloom et al. (2016) Fig. 3) can be interpreted as proper ages. In this respect it also interesting to see that shortly in the future this doesn't work any more. For your paper this would mean that you drastically shorten it by dropping anything else (i.e. in particular the simulations S1, S2 and all stuff relating to the separation of transit time into contributions from ACC and MAC). With such changes I think a resubmission could make sense.

**Minor Remarks**

- For a resubmission a better polished text would be appreciated. The current text is mostly understandable but the English shows quite some deficits (an annoyingly plentitude of missing articles; wrong grammar (lines 41, 47, 127, 154, 156, 175, 177, 178, 261, 357, 479; Supplement scattered with errors); incomplete formulations (lines 56, 139, 167, 210); wrong or missing preposition (lines 99, 333), ununderstandable formulations (line 168, 196, 274)). And results should be presented in present tense not in past tense as e.g. in the abstract.
- I do not see why in addition to the terms 'transit time' and 'turnover time' one needs the equivalent use of namings 'Olson method' and 'Rasmussen method', respectively, this is only confusing – by whatever method you compute turnover or transit time,

they remain the same.

- In the model description for CABLE you refer for the photosynthesis part to a paper by Farquhar that refers to the C3 pathway only. What does it mean for the realism of your simulations that CABLE is not accounting for C4 photosynthesis, happening at huge areas wordwide?
- What about land use change? This seems to be not accounted for in CABLE, but replacement of forests with agricultural lands could in principle speed up the land carbon cycle by one magnitude (maybe forests: 30 years vs. agriculture 1 year).
- What about natural vegetation? How does it change in your CABLE simulations?
- For the historical period CABLE was forced by CRU-NCEP data, while for 2006-2100 simulation data from CESM were used. How good do these simulated climate data fit at the transition period around 2005/2006 to the historical values, concerning e.g. the global and zonal levels of land temperature, precipitation, and radiation?
- When you introduce the Rasmussen method to calculate transit time it would be good to mention that this approach works only for linear box models. Is CABLE really of this type? You could demonstrate this by listing in the appendix the box-model equations for CABLE (like in section 8 of the Rasmussen et al. paper) – this would also help to make precise what at all your mathematical symbols mean. – I wonder about the applicability of the Rasmussen et al. approach because I would think that e.g. the phenology introduces some non-linearity in the dynamics of leaf carbon since leaf area cannot grow beyond a certain value depending on vegetation type – but maybe CABLE works differently. And what about structural allometries between different plant parts (stems, roots, leaves) that are also non-linear?
- Fig. 1: (i) Title Fig. 1a: Transient$\rightarrow$ Transit. (ii) Make the scale numbering for both plots of Fig. 1 better readable (e.g. in steps of 5 or 10 years, or, if logarithmic, use other round numbers, but definitely not something like 3623). (iii) Are the colors at the edges of e.g. Antarctica and Greenland really a result of your simulations, or is it a plotting artefact e.g. from your grid cell interpolations?
- Lines 147-150: I guess that this paragraph should say that the authors solve equation (2) by an Euler method starting from zero land carbon – this should be stated more clearly.
- Fig. 2: Why don't you also plot turnover time from CABLE? This would make it even more clear that transient and turnover time match well for this period of time.
- Why do you talk of "permafrost areas" instead of e.g. "high latitudes"? I guess that CABLE is not accounting for permafrost.
- Fig. 6g: You attribute the small difference in turnover and transit time for the stationary state to the presence of the seasonal cycle that makes the system non-stationary. Can this explain the increase of this difference beyond 60°N?

**Literature**

A.A. Bloom et al. *The decadal state of the terrestrial carbon cycle: global retrievals of terrestrial carbon allocation, pools, and residence times*, PNAS 113 (2016) 12851290.

---

## Referee Comment (RC2) · Anonymous Referee #2 · 23 Jun 2018

In this manuscript, Lu and colleagues use the CABLE model to show: a) how turnover time and transit time diverge under transient global change simulations, and b) decompose the contribution of turnover time between the age structure of ecosystem pools and their contribution to the output flux. This is an exciting and important paper. Previous studies have shown how turnover time contributes to our predictive uncertainty of the future response of the terrestrial biosphere to global change (e.g. Friend et al., 2013). However, this study nicely shows that turnover times themselves can also be an uncertain metric to assess model performance and quantify carbon storage potential in the terrestrial biosphere under non steady-state conditions.

[Figure]

The manuscript expands on previous work by Rasmussen et al. (2016) who developed formulas for the mean transit time for non-steady-state conditions. It shows how global change drivers such as warming and $CO_2$ can modify the time that carbon requires to transit through the terrestrial biosphere. The implications are not only for comparing two different modeling metrics, but it helps to understand how global change modifies the time scales of carbon storage in the terrestrial biosphere.

Unfortunately, the manuscript has problems with the English language (typos, grammar), but if these issues are addressed with the help of a native English speaker, the manuscript can be published with minor revisions. I only have a few minor comments to help improve the manuscript:

- Line 22. Increase with respect to what? Do you mean increase in the transient simulations with respect to steady-state? Please clarify.

- Line 29 plus 3 other occurrences. Change Olsen to Olson.

- Figure 2. I don't understand why you plot together the turnover times from Carvalhais et al. (2014) versus the dynamic transit times. They are conceptually different and computed in very different ways. This figure gives the false impression that these metrics should be compared, and that they are roughly equal, which this very same manuscript clearly shows that they are not. I suggest removing this figure to avoid confusion.

**References**

Carvalhais, N., Forkel, M., Khomik, M., Bellarby, J., Jung, M., Migliavacca, M., Mu, M., Saatchi, S., Santoro, M., Thurner, M., et al. (2014). Global covariation of carbon turnover times with climate in terrestrial ecosystems. *Nature*, 514(7521):213.

[Figure]

Friend, A. D., Lucht, W., Rademacher, T. T., Keribin, R., Betts, R., Cadule, P., Ciais, P., Clark, D. B., Dankers, R., Falloon, P. D., Ito, A., Kahana, R., Kleidon, A., Lomas, M. R., Nishina, K., Ostberg, S., Pavlick, R., Peylin, P., Schaphoff, S., Vuichard, N., Warszawski, L., Wiltshire, A., and Woodward, F. I. (2013). Carbon residence time dominates uncertainty in terrestrial vegetation responses to future climate and atmospheric CO2. *Proceedings of the National Academy of Sciences*, 111(9):3280–3285.

Rasmussen, M., Hastings, A., Smith, M. J., Agusto, F. B., Chen-Charpentier, B. M., Hoffman, F. M., Jiang, J., Todd-Brown, K. E. O., Wang, Y., Wang, Y.-P., and Luo, Y. (2016). Transit times and mean ages for nonautonomous and autonomous compartmental systems. *Journal of Mathematical Biology*, 73(6):1379–1398.
* * *

---

## Author Comment (AC1) · 28 Jul 2018

Dear Referee 2:

We are very appreciated your comments on our manuscript. We have carefully read your comments. Hopefully, you will find our response satisfactory.

Xingjie Lu

On Behalf of all co-authors

Reviewer 2: Lu and colleagues use the CABLE model to show: a) how turnover time and transit time diverge under transient global change simulations, and b) decompose

the contribution of turnover time between the age structure of ecosystem pools and their contribution to the output flux. This is an exciting and important paper. Previous studies have shown how turnover time contributes to our predictive uncertainty of the future response of the terrestrial biosphere to global change (e.g. Friend et al., 2013). However, this study nicely shows that turnover times themselves can also be an uncertain metric to assess model performance and quantify carbon storage potential in the terrestrial biosphere under non steady-state conditions. The manuscript expands on previous work by Rasmussen et al. (2016) who developed formulas for the mean transit time for non-steady-state conditions. It shows how global change drivers such as warming and CO2 can modify the time that carbon requires to transit through the terrestrial biosphere. The implications are not only for comparing two different modeling metrics, but it helps to understand how global change modifies the time scales of carbon storage in the terrestrial biosphere.

Response: Thanks for the positive comments on our manuscript.

Reviewer 2: Unfortunately, the manuscript has problems with the English language (typos, grammar), but if these issues are addressed with the help of a native English speaker, the manuscript can be published with minor revisions. I only have a few minor comments to help improve the manuscript:

Response: Thanks for the suggestion. We will find an English native speaker to help edit the language.

Reviewer 2: Line 22. Increase with respect to what? Do you mean increase in the transient simulations with respect to steady-state? Please clarify.

Response: Yes, increase with respect to steady state. We will revise the sentence to be clearer.

Reviewer 2: Line 29 plus 3 other occurrences. Change Olsen to Olson.

Response: We will revise all of them as suggested. Sorry for the typos.

Reviewer 2: Figure 2. I don't understand why you plot together the turnover times from Carvalhais et al. (2014) versus the dynamic transit times. They are conceptually different and computed in very different ways. This figure gives the false impression that these metrics should be compared, and that they are roughly equal, which this very same manuscript clearly shows that they are not. I suggest removing this figure to avoid confusion.

Response: Thanks for pointing out the confusions we might have made without enough details. We agree that turnover time and transit time are calculated in different ways. However, theoretically, turnover time and transit time should be strictly equal under steady state condition (Sierra et al., 2017). Our assumption, which is also used by some other studies, is that ecosystem C cycle may be close to the steady state in present-day, however, climate change may drive C cycle to a non-steady state in the future. Therefore, C transit time is comparable with C turnover time at present-day in Fig. 2. This figure serves as a validation of our model against the observations, which is very important for a modeling study. More importantly, reviewer 1 really likes it. As such, we would keep Figure 2, but will add more details, e.g., our assumption, in the figure caption and will change "Rasmussen method" to "model simulations" to avoid any confusions.

References:

Sierra, C. A., Muller, M., Metzler, H., Manzoni, S., and Trumbore, S. E.: The muddle of ages, turnover, transit, and residence times in the carbon cycle, Global Change Biol, 23, 1763-1773, 2017.

---

## Author Comment (AC3) · 13 Sep 2018

Dear Referee 1:

We greatly appreciate your time and effort to read, understand, and make comments on our manuscript. We have carefully studied your comments. Hope our responses have adequately addressed your concerns.

Xingjie Lu and Yiqi Luo

On behalf of all co-authors

Anonymous Referee #1

Reviewer 1: It is well known that carbon turnover time as computed by carbon stock divided by carbon flux has a well defined meaning only for stationary states, while the meaning of transit time as mean age of carbon released from the system remains valid also for nonstationary states. On this background the submitted paper investigates how strongly transit and turnover times deviate in historical and RCP8.5 scenario simulations. The simulations are performed using the land surface model CABLE in offline simulations forced by CRU-NCEP (historical) and CLM (scenario) data. To separate the physical and biogeochemical effect of CO2 on transit and turnover times, separate simulations are performed where either the temperature forcing or the photosynthetically relevant CO2 level are kept fixed. To determine transit time using the approach by Rasmussen et al. (2016), the authors equipped CABLE with a diagnostic to follow changes in carbon stocks of all pools of their land carbon model and the fluxes between them. The authors show that the expression for changes in transit time based on this approach can be separated into two components, one arising from changes in the mean age of the carbon in the different pools (abbreviated in the following as MAC – Mean Age Change), the other arising from changes in the age composition of the carbon fluxes 'respired' from the different carbon pools (abbreviated in the following as ACC – Age Composition Change). The authors show how the MAC and ACC contributions to transit time change in their scenario simulations.

Response: We greatly appreciate the reviewer for carefully reading our manuscript. The above paragraph is a good summary of what we did in our study.

Major Remarks

Reviewer 1: 1) The study is not well motivated

In the abstract the authors motivate their study by writing that considering transit and turnover times "neither of them has been carefully examined under transient C dynamics in response to climate change". This is not a very convincing argument for their study since (i) the study should not be published if its contents would not be new, and

(ii) that something hasn't been done doesn't qualify it as scientifically relevant.

Response: We thank the reviewer for the great point. First, we totally agree that "something hasn't been done does not mean it is scientifically relevant". The reviewer is a critical thinker. He or she may agree that any sentence out of a context may not make sense in a manuscript. A sentence, however, carries meanings in connection with other sentences. For our manuscript, the first sentence of the abstract is "Ecosystem carbon (C) transit time is a critical diagnostic parameter to characterize land C sequestration." Combining the first sentence with the third sentence in the abstract can form a sentence "Such a critical parameter 'has not been carefully examined under transient C dynamics in response to climate change.'" This new sentence by placing "not been carefully examined" in the context, we think, identifies an important knowledge gap and thus makes our study scientifically relevant. Nevertheless, we take the criticism seriously from the reviewer and changes the sentence to be "However, we know little about whether transit time or turnover time is a better diagnostic parameter to represent carbon cycling through multiple pools under non steady state."

We also agree with reviewer's statement "the study should not be published if its contents would not be new". Since the reviewer did not elaborate this point in reference to our manuscript, we consider her or his comment is a general statement.

Reviewer 1: And also the introduction is not clear about the motivation of the study, except that from the subtext the authors let arise the impression that results from other studies using turnover time for a non-steady state situation cannot be trusted. Here e.g. a study by He et al. (2016) is cited with the result that in CMIP5 simulations the "soil C sequestration potential can be overestimated due to under estimation of C turnover time". But in this study "turnover times" are decay parameters of a model and not a diagnostic turnover time computed by carbon stock divided by carbon flux so that this study is not suitable for motivating the study reviewed here.

Response: We greatly appreciate the reviewer's effort to identify the motivation of our
study in the introduction section. The last sentence in the first paragraph of the intro-
duction section states: "It is not clear how much estimates of C turnover time deviate
from mean C transit time and what cause their deviation under climate change." If this
is not clear to the reviewer, we were wondering if the reviewer has any more specific
suggestion to revise our sentence so as to make our motivation of the study apparent
to her/him.

The reviewer is very knowledgeable and knows the technical detail very well in the
study by He et al. (2016). The reviewer is right that in He et al. (2016) study, "turnover
times" were derived from decay parameters of a model and not by carbon stock divided
by carbon flux. Our sentence "Up to 40% of soil C sequestration potential can be
overestimated due to underestimation of C turnover time in current CMIP5 models (He
et al., 2016)" reflects a main conclusion from that study. Besides, He et al. (2016)
used the term "turnover time" in their paper. As Sierra et al. (2017) pointed out, there
are many different ways to define "turnover time" in the literature. We used the study
by He et al. (2016) as one example to highlight the need to understand turnover time
better instead of to define the term of turnover time. We were puzzled why the reviewer
thought the paper by He et al. (2016) "is not suitable for motivating" our study.

Reviewer 1: A similar remark concerns the study of Friend et al. (2013) cited in the
introduction: Friend et al. indeed calculate turnover time as carbon stock divided by
carbon flux but as a diagnostic to test the validity of their simple carbon model, but not
for drawing any other conclusions from it that could be improved by using transit time.
Hence, in my opinion the authors handle the cited literature inappropriately to motivate
their study and thereby give a wrong impression of their relevance.

Response: The reviewer used another sentence in the manuscript to question the mo-
tivation of our study. The sentence "Recent model inter-comparison study indicated
that a major cause of uncertainty in predicting future terrestrial C sequestration is the
variation in C turnover time among the models (Friend et al., 2014)" was used to high-
light the need to understand turnover time better as well. We were very confused by

reviewer's point that because Friend et al. (2014) did not mention transit time in their paper, we could not use it to motive our study on turnover time in relation with transit time.

Overall, we are grateful to the reviewer for his/her critiques, which made us to carefully examine our manuscript again.

Reviewer 1: 2) The relevance of the results of the study is unclear

According to section 5 ('Conclusions') the study has two major results. The first (lines 357-361) concerns the development of transit and turnover time in their simulations, in particular that they increasingly deviate from one another during the 21st century, and that the deviations are stronger in some regions than in others. But what to conclude from this? Is this a useful knowledge?

Response: We were not sure if the reviewer asked those two questions philosophically or practically. In practice, knowledge about the deviation between C transit time and turnover time in different regions under different scenarios (warming and [CO2] rising) is useful for us to understand time characteristic of the ecosystem carbon dynamics. When we lump all pools and fluxes together to calculate turnover time by "stock over flux", the time characteristic is different from that of transit time when individual pools and fluxes are considered to be networked together to form compartmental dynamical system. Thus, our results provides information on how turnover time deviates from transit time in specific region and ecosystems. Because "stock over flux" is still the easiest way to measure how fast C cycle through ecosystem, it is not our purpose to persuade the community to completely give up the method. Instead, we only provide information about how carbon transit time deviates with the turnover time in the non-steady state. We estimated both C transit time and turnover time globally. We assume the deviation between them is due to the lost information by lumping pools and fluxes to calculate the turnover time. The temporal and spatial estimates on the deviation tell us whether we can still use turnover time at a specific place and time.

If those are philosophical questions, we may not be able to answer either of the questions correctly. In this case, it is only reviewer's perspective that matters.

Reviewer 1: The second result (lines 363-368) is that transit time can be separated into contributions from MAC and ACC because the residual is small (see eq. (6) and Fig. 3). Therefore on first sight I indeed thought that this separation is an interesting idea that could help to understand better how transit time behaves under different forcings. The authors claim (lines 364/365) that MAC is determined by carbon input and ACC by "differential responses of various C pools to climate warming and rising atmospheric [CO2]". While this latter formulation is rather cryptic, I take from it that the authors think that one could pin down what affects MAC and ACC so that e.g. for scenarios of different CO2 and/or temperature rise one could understand why transit time would develop differently.

Response: We thank the reviewer for her/his carefully examining the separation of Mean Age Change (MAC) from Age Composition Change (ACC) in our study. We agree that the contributions by MAC and ACC are not independent with each other, similarly as pool and flux influence each other. But the separation of pool and flux is so fundamental for any research to understand carbon cycle. In this study MAC is related to the change caused by carbon age, which is equivalent to pool change in the pool-flux separation whereas ACC is more about the change caused by flux.

We understand that it may take time to fully comprehend new concepts about MAC and ACC. The separation was made according to derivative of an equation with two components. This method is very commonly used in many studies. For example, Koven et al. (2015) used this method to separate relative contributions of NPP vs. turnover time in influencing carbon sequestration. By separating the two terms, we can better understand mechanisms underlying the changes in transit time.

Reviewer 1: But I very much doubt that this separation helps understanding anything since there is no way to see how MAC and ACC are separately affected by the carbon

inputs or the forcings: this is because MAC and ACC are not independent from one another since they are both derived from the same development of carbon stored in the compartments (xi(t) in eq. (2) and eq. line 166). Hence a change of the carbon input into the system changes both MAC and ACC, and also a change in pool turnover time parameters by a changing climate (temperature, moisture) changes both MAC and ACC.

Response: The reviewer "very much doubt that this separation helps understanding anything since there is no way to see how MAC and ACC are separately affected by the carbon inputs or the forcings". We do not agree that contributions from different reasons cannot be separated only because they are both partly affected by the same factor. There are many opposite examples in C cycle study. For examples, C pool and flux may both respond to change in C input. However, their responses have different ecological meanings.

Moreover, eq. (6) is the major equation to show the differences between MAC and ACC. MAC is mainly contributed by the change in age and ACC is mainly contributed by the change in composition.

Reviewer 1: Only if one could understand how climate and CO2 act differently on MAC and ACC this separation could contribute to a better understanding of transit time development in transient simulations.

Response: Reviewer was wondering "how climate and CO2 act differently on MAC and ACC". Our results have shown different responses of MAC and ACC to climate and CO2. Figure 3d shows that contribution from ACC (in blue) changes from negative to positive in response to C balance change, whereas MAC (in red) keep increasing. These features are very similar to C pool and flux that C flux response faster than C pool to input change. More evidence that contributions from MAC and ACC vary spatially (Figure 4) also indicates the separation is helpful for our understanding.

Reviewer 1: That MAC and ACC have no individual meaning can also be seen directly

in the simulation results: In the simulation with both forcings combined (simulation S3) the contribution to transit time from ACC is not even approximately the sum of the ACC contributions from the simulations with forcings separated (S1, S2), and the same is true for MAC. Hence, the behaviour of simulation S3 cannot be understood as combination of results from S1 and S2.

Response: The reviewer also doubted that "MAC and ACC have no individual meaning" because the sum of individual effects (warming effect and [CO2] rising effect) does not equal to the combined effects. The non-additive MAC or ACC in response to warming and [CO2] rising is possibly due to the non-linear or interactive effects, which have commonly been found in other studies. For examples, climate and rising atmospheric [CO2] affect GPP together, but usually their co-effects are not equal to the sum of their individual effects (Zhang et al., 2016). However, it does not necessarily mean climate effects and rising atmospheric [CO2] effects on GPP should not be separated.

Meanwhile, MAC and ACC do have individual meaning. In theory, Eqn (6) has illustrated that MAC represents the contribution from change in age and ACC represents the contribution from change in composition. In practice, results from last response (Figure 3d and Figure 4) have also confirmed they are completely different.

Reviewer 1: – In conclusion, I think this separation is only technical and pretty useless. In order to convince me from the opposite, the authors had to show me a case where it leads to an improved understanting.

Response: In practice, the contributions of MAC and ACC should be different among models. None of previous studies have diagnosed those two contributions separately. Although total change in C transit time has been compared recently (Sierra, 2017), a thorough assessment on their individual contributions would provide more useful information. Otherwise, there is still a chance for models to get the right answer with wrong reasons. By combining compartment models with different types of measurements via data assimilation techniques, we may be able to better constrain MAC and ACC respectively. Therefore, modelled C cycle can be better calibrated by constraining MAC and ACC against measurements in the future.

Reviewer 1: Concernig the other remarks related to the second result in lines 366-368, I think they are all wrong: (i) The calculation of turnover time by dividing stocks by fluxes is not assuming anything, it is simply a diagnostic that in the case of stationary states has a well defined meaning, but can, as a diagnostic, still be a useful concept (see my remark on the study by Friend et al. (2013) above).

Response: We appreciate that this reviewer clearly shows her/his view and perspective. He or she stand strongly for using turnover time. We agree that turnover time is a good and useful diagnostic, since both stock and fluxes are easy to measure. Our manuscript have admitted the advantages of C turnover time "C turnover time can be easily calculated from C stock over flux, both of which can be easily measured." (Line 342-343) However, would his/her statement "the calculation of turnover time is not assuming anything" be contradictory to "it is simply a diagnostic that in the case of stationary states has a well defined meaning"? Would "in the case of stationary states" be an assumption? But we hope the reviewer agrees that turnover time is calculated by lumping pools and fluxes together. As a scientist, he or she, we hope, will not be against research to explore other ideas related to time characteristics of carbon cycle.

Reviewer 1: (ii) Surely turnover time changes when MAC and ACC change, so that contrary to the authors claim it accounts for such changes.

Response: We mentioned "C turnover time does not account for changes in age structure and contribution fractions of different pools to ecosystem respiration." (Line 366-367). This sentence did not say "turnover time does not change". Obviously, C turnover changes when MAC and ACC changes. However, we care about whether the diagnostic really accounts for the certain critical information. The "change" in C turnover time with MAC and ACC is not really sufficient enough to accurately quantify the contributions from MAC and ACC to C transit time.

Reviewer 1: Hence (iii) contrary to the claim by the authors one cannot conclude that transit time is a "better parameter". – A similar claim is found in the last scentence of the abstract where the conclusion is even weirder by saying that the use of turnover time instead of transient time may "lead to biases in estimating land C sequestration" – how could the mere calculation of a time scale affect the estimation of C sequestration?

Response: The reviewer seems to believe that there is no better diagnostic than turnover time. We partly agree but it should depend on the specific case we are studying. Eg. Friend et al., (2014) identified that the source of the uncertainty in predicating land C sequestration C is mainly from turnover time. Qualitatively, turnover time and transit time behave similarly. Thus, the turnover time can be used in this case to point out a direction for model improvement. However, turnover time may not represent the time characteristic of carbon dynamics if we are interested in carbon sequestration in multiple pools. It is because turnover time uses lumped pools and fluxes for calculation.

Reviewer 1: 3) Some suggestions for improving the paper

There is one result of the paper – surprisingly not mentioned in the conclusions – that in my opinion makes an important contribution to land carbon research: This is the comparison of the CABLE results for transit time with the observational results by Carvalhais et al. (2014) in Fig. 2. When the Carvalhais et al. paper appeared, I thought it's nice that they produced a map of stocks divided by fluxes so that this turnover time can be used as a diagnostic to easily compare with results from model simulations. But with the study under review here, we now know that despite the non-stationarity of todays carbon cycle, turnover times agree well with transit times (Fig. (5)) so that the observational turnover times of Carvalhais et al. (2014) can indeed be interpreted as proper carbon ages. And that the zonal distribution of CABLE results matches those of Carvalhais et al. quite well provides additional credit to this conclusion.

Response: Thanks for the great suggestion to improve our manuscript. We will include this point in the discussions to support Carvalhais et al. (2014).

Reviewer 1: Hence, what I propose is that you focus your study on the question to what extend the observational estimates of turnover time by Carvalhais et al. (2014) (and if possible also those by Bloom et al. (2016) Fig. 3) can be interpreted as proper ages. In this respect it also interesting to see that shortly in the future this doesn't work any more. For your paper this would mean that you drastically shorten it by dropping anything else (i.e. in particular the simulations S1, S2 and all stuff relating to the separation of transit time into contributions from ACC and MAC). With such changes I think a resubmission could make sense.

Response: Although this reviewer clearly has her/his own preference, these are very constructive suggestions. We fully understand the reviewer's point and do see some interesting conclusions being drawn in this way. It is very interesting that the reviewer states "In this respect it also interesting to see that shortly in the future this doesn't work any more." We thought we just did what the reviewer suggested us to do. Our analysis indicates that turnover time works quite well now and until the middle of this century before it significantly deviates from transit time. However, to do what the reviewer suggested us to do, we need the full length of the manuscript. Anyway, we will certainly revise the manuscript to address reviewer's comments.

Minor Remarks

Reviewer 1: For a resubmission a better polished text would be appreciated. The current text is mostly understandable but the English shows quite some deficits (an annoyingly plentitude of missing articles; wrong grammar (lines 41, 47, 127, 154, 156, 175, 177, 178, 261, 357, 479; Supplement scattered with errors); incomplete formulations (lines 56, 139, 167, 210); wrong or missing preposition (lines 99, 333), ununderstandable formulations (line 168, 196, 274)). And results should be presented in present tense not in past tense as e.g. in the abstract.

Response: We will carefully check grammar, formulations, etc. and will use present tense. We will have a native English speaker help polish the whole text.

[Figure]

Reviewer 1: I do not see why in addition to the terms 'transit time' and 'turnover time' one needs the equivalent use of namings 'Olson method' and 'Rasmussen method', respectively, this is only confusing – by whatever method you compute turnover or transit time, they remain the same.

Response: Thanks for pointing out the confusion. In the revision, we will use only transit time and turnover time avoid the confusion.

Reviewer 1: In the model description for CABLE you refer for the photosynthesis part to a paper by Farquhar that refers to the C3 pathway only. What does it mean for the realism of your simulations that CABLE is not accounting for C4 photosynthesis, happening at huge areas wordwide?

Response: Sorry for the misleading, but CABLE does have C4 pathway photosynthesis as well, which follows the method by Kowalczyk et al. (2006). In the revision, we will add this information to clarify.

Reviewer 1: What about land use change? This seems to be not accounted for in CABLE, but replacement of forests with agricultural lands could in principle speed up the land carbon cycle by one magnitude (maybe forests: 30 years vs. agriculture 1 year).

Response: Land use change is not accounted for in CABLE and we have discussed the possible bias caused by this in the discussion section 4.3.

Reviewer 1: What about natural vegetation? How does it change in your CABLE simulations?

Response: CABLE is not a dynamics vegetation model, which means natural vegetation distribution is static. In the revision, we will add some discussions on how changes in natural vegetation distribution in response to climate change will influence the estimates of transit time.

Reviewer 1: For the historical period CABLE was forced by CRU-NCEP data, while

for 2006-2100 simulation data from CESM were used. How good do these simulated climate data fit at the transition period around 2005/2006 to the historical values, concerning e.g. the global and zonal levels of land temperature, precipitation, and radiation?

Response: We have realized this and adjusted the CESM future simulation results to make the connection between historical climate and RCP8.5 smooth and also to make the trends and seasonal and daily variability in the climate data simulated by CESM follow their patterns in the historical data in CRU-NCEP. For example, Figure R1 shows how global mean annual temperature changes before and after being adjusted. In the revision, we will add these figures in the appendix.

Fig. R1 Global mean annual temperature changes from 1901 to 2100. Historical data were interpolated from 6-hourly (Qian et al., 2006) to hourly and re-gridded from a spatial resolution of 0.5o by 0.5o to 1.875o by 2.5o. From 2006 to 2100, the hourly meteorological variables were generated by Community Earth System Model version 1.0 (CESM) (Li et al., 2016; Hurrell et al., 2013) for Representative Concentration Pathway (RCP) 8.5. The red line from 2006-2100 represents original model results and the blue line represents the global mean annual temperature being adjusted to make the connection between historical climate and RCP8.5 smooth and also to fit the trends and seasonal and daily variability in the historical data.

Reviewer 1: When you introduce the Rasmussen method to calculate transit time it would be good to mention that this approach works only for linear box models. Is CABLE really of this type? You could demonstrate this by listing in the appendix the box-model equations for CABLE (like in section 8 of the Rasmussen et al. paper) – this would also help to make precise what at all your mathematical symbols mean. – I wonder about the applicability of the Rasmussen et al. approach because I would think that e.g. the phenology introduces some non-linearity in the dynamics of leaf carbon since leaf area cannot grow beyond a certain value depending on vegetation type – but maybe CABLE works differently. And what about structural allometries between

different plant parts (stems, roots, leaves) that are also non-linear?

Response: These questions and suggestions are very helpful to improve the manuscript. We will give more details about CABLE in appendix. Yes, C cycle in CABLE, even with phenology processes, can be considered as a linear model. In the deciduous plant functional type, CABLE's phenology only changes the leaf turnover rate and allocation fraction in spring and fall. When LAI grows over the upper limit, CABLE will set the allocation fraction to leaf to "0" in order to prevent further growth of leaf. The dynamics of both turnover rate and allocation fraction are determined by time-dependent environmental scalars. Because the environmental scalars are independent from C pool sizes in most cases, the model can be considered as a linear model. In addition, CABLE does not include the structural allometries. The C pool size only depends on the input (net primary production allocated) and output (turnover) and in most of time, the allocation fraction is independent from C pool sizes.

In the revision, we will add all these details in the appendix.

Reviewer 1: Fig. 1: (i) Title Fig. 1a: Transient→ Transit. (ii) Make the scale numbering for both plots of Fig. 1 better readable (e.g. in steps of 5 or 10 years, or, if logarithmic, use other round numbers, but definitely not something like 3623).

Response: We will make revisions as suggested.

Reviewer 1: (iii) Are the colors at the edges of e.g. Antarctica and Greenland really a result of your simulations, or is it a plotting artefact e.g. from your grid cell interpolations?

Response: Those colors at the edges are the real results of the simulations. The red indicates that C transit time and mean age are really high in the high latitude region. In contrast, the edges of islands at lower latitudes, e.g., Hawaii, are relative low.

Reviewer 1: Lines 147-150: I guess that this paragraph should say that the authors solve equation (2) by an Euler method starting from zero land carbon – this should be

stated more clearly.

Response: Yes, we will follow the suggestion to describe more clearly.

Reviewer 1: Fig. 2: Why don't you also plot turnover time from CABLE? This would make it even more clear that transient and turnover time match well for this period of time.

Response: Thanks for the great suggestion. We will add turnover time in Figure 2.

Reviewer 1: Why do you talk of "permafrost areas" instead of e.g. "high latitudes"? I guess that CABLE is not accounting for permafrost.

Response: We agree that "high latitudes" is more accurate. We will change the wording.

Reviewer 1: Fig. 6g: You attribute the small difference in turnover and transit time for the stationary state to the presence of the seasonal cycle that makes the system non-stationary. Can this explain the increase of this difference beyond 60âŮęN?

Response: Yes, conceptually, the significant bias in high latitude should be due to the seasonal cycle. We will illustrate it in the results.

References:

Carvalhais, N., Forkel, M., Khomik, M., Bellarby, J., Jung, M., Migliavacca, M., Mu, M. Q., Saatchi, S., Santoro, M., Thurner, M., Weber, U., Ahrens, B., Beer, C., Cescatti, A., Randerson, J. T., and Reichstein, M.: Global covariation of carbon turnover times with climate in terrestrial ecosystems, Nature, 514, 213-+, 10.1038/nature13731, 2014.

Friend, A. D., Lucht, W., Rademacher, T. T., Keribin, R., Betts, R., Cadule, P., Ciais, P., Clark, D. B., Dankers, R., Falloon, P. D., Ito, A., Kahana, R., Kleidon, A., Lomas, M. R., Nishina, K., Ostberg, S., Pavlick, R., Peylin, P., Schaphoff, S., Vuichard, N., Warszawski, L., Wiltshire, A., and Woodward, F. I.: Carbon residence time dominates uncertainty in terrestrial vegetation responses to future climate and atmospheric CO2,

P Natl Acad Sci USA, 111, 3280-3285, 10.1073/pnas.1222477110, 2014.

He, Y. J., Trumbore, S. E., Torn, M. S., Harden, J. W., Vaughn, L. J. S., Allison, S. D., and Randerson, J. T.: Radiocarbon constraints imply reduced carbon uptake by soils during the 21st century, Science, 353, 1419-1424, 10.1126/science.aad4273, 2016.

Hurrell, J. W., Holland, M. M., Gent, P. R., Ghan, S., Kay, J. E., Kushner, P. J., Lamarque, J. F., Large, W. G., Lawrence, D., Lindsay, K., Lipscomb, W. H., Long, M. C., Mahowald, N., Marsh, D. R., Neale, R. B., Rasch, P., Vavrus, S., Vertenstein, M., Bader, D., Collins, W. D., Hack, J. J., Kiehl, J., and Marshall, S.: The Community Earth System Model A Framework for Collaborative Research, B Am Meteorol Soc, 94, 1339-1360, 10.1175/Bams-D-12-00121.1, 2013.

Koven, C. D., Chambers, J. Q., Georgiou, K., Knox, R., Negron-Juarez, R., Riley, W. J., Arora, V. K., Brovkin, V., Friedlingstein, P., and Jones, C. D.: Controls on terrestrial carbon feedbacks by productivity versus turnover in the CMIP5 Earth System Models, Biogeosciences, 12, 5211-5228, 2015.

Li, J. D., Wang, Y. P., Duan, Q. Y., Lu, X. J., Pak, B., Wiltshire, A., Robertson, E., and Ziehn, T.: Quantification and attribution of errors in the simulated annual gross primary production and latent heat fluxes by two global land surface models, J Adv Model Earth Sy, 8, 1270-1288, 10.1002/2015ms000583, 2016.

Qian, T. T., Dai, A., Trenberth, K. E., and Oleson, K. W.: Simulation of global land surface conditions from 1948 to 2004. Part I: Forcing data and evaluations, J Hydrometeorol, 7, 953-975, Doi 10.1175/Jhm540.1, 2006.

Sierra, C.: Soil organic matter persistence as a stochastic process: age and transit time distributions of carbon in soils, EGU General Assembly Conference, Vienna, Austria, 2017.

Sierra, C. A., Muller, M., Metzler, H., Manzoni, S., and Trumbore, S. E.: The muddle of ages, turnover, transit, and residence times in the carbon cycle, Global Change Biol,

23, 1763-1773, 2017.

[Figure]

[Figure]

**Fig. 1.** Fig. R1 Global mean annual temperature changes from 1901 to 2100. Historical data were interpolated from 6-hourly (Qian et al., 2006) to hourly and re-gridded from a spatial resolution of 0.5o by 0.5

---

## Author Response (AR1)

**Dear Referee 1:**

We greatly appreciate your time and effort to read, understand, and make comments on our manuscript. We have carefully studied your comments. Hope our responses (in blue) have adequately addressed your concerns.

5 Xingjie Lu and Yiqi Luo On behalf of all co-authors

**Anonymous Referee #1**

Reviewer 1: It is well known that carbon turnover time as computed by carbon stock divided by carbon flux has a well defined

- 10 meaning only for stationary states, while the meaning of transit time as mean age of carbon released from the system remains valid also for nonstationary states. On this background the submitted paper investigates how strongly transit and turnover times deviate in historical and RCP8.5 scenario simulations. The simulations are performed using the land surface model CABLE in offline simulations forced by CRU-NCEP (historical) and CLM (scenario) data. To separate the physical and biogeochemical effect of CO2 on transit and turnover times, separate simulations are performed where either the temperature forcing or the
- 15 photosynthethically relevant CO2 level are kept fixed. To determine transit time using the approach by Rasmussen et al. (2016), the authors equipped CABLE with a diagnostic to follow changes in carbon stocks of all pools of their land carbon model and the fluxes between them. The authors show that the expression for changes in transit time based on this approach can be separated into two components, one arising from changes in the mean age of the carbon in the different pools (abbreviated in the following as MAC Mean Age Change), the other arising from changes in the age composition of the carbon fluxes
- 20 'respired' from the different carbon pools (abbreviated in the following as ACC Age Composition Change). The authors show how the MAC and ACC contributions to transit time change in their scenario simulations.

Response: We greatly appreciate the reviewer for carefully reading our manuscript. The above paragraph is a good summary of what we did in our study.

**25 Major Remarks**

**Reviewer 1: 1) The study is not well motivated**

In the abstract the authors motivate their study by writing that considering transit and turnover times "neither of them has been carefully examined under transient C dynamics in response to climate change". This is not a very convincing argument for their study since (i) the study should not be published if its contents would not be new,

30 Response: We thank the reviewer for the great point. First, we totally agree that "something hasn't been done does not mean it is scientifically relevant". The reviewer is a critical thinker. He or she may agree that any sentence out of a context may not make sense in a manuscript. A sentence, however, carries meanings in connection with other sentences. For our manuscript, the first sentence of the abstract is "Ecosystem carbon (C) transit time is a critical diagnostic parameter to characterize land C sequestration." Combining the first sentence with the third sentence in the abstract can form a sentence "Such a critical

35 parameter 'has not been carefully examined under transient C dynamics in response to climate change."" This new sentence by placing "not been carefully examined" in the context, we think, identifies an important knowledge gap and thus makes our study scientifically relevant.

Nevertheless, we take the criticism seriously from the reviewer and changes the sentence to be "However, we know little about whether transit time or turnover time better represents carbon cycling through multiple compartments under non steady state." (See Line 16-18)

40

Reviewer 1: and (ii) that something hasn't been done doesn't qualify it as scientifically relevant.

Response: We also agree with reviewer's statement "the study should not be published if its contents would not be new". Since the reviewer did not elaborate this point in reference to our manuscript, we consider her or his comment is a general statement. Reviewer 1: And also the introduction is not clear about the motivation of the study, except that from the subtext the authors

- 45 let arise the impression that results from other studies using turnover time for a non-steady state situation cannot be trusted. Here e.g. a study by He et al. (2016) is cited with the result that in CMIP5 simulations the "soil C sequestration potential can be overestimated due to under estimation of C turnover time". But in this study "turnover times" are decay parameters of a model and not a diagnostic turnover time computed by carbon stock divided by carbon flux so that this study is not suitable for motivating the study reviewed here.
- 50 Response: We greatly appreciate the reviewer's effort to identify the motivation of our study in the introduction section. The last sentence in the first paragraph of the introduction section states: "It is not clear how much estimates of C turnover time deviate from mean C transit time and what cause their deviation under climate change." If this is not clear to the reviewer, we were wondering if the reviewer has any more specific suggestion to revise our sentence so as to make our motivation of the study apparent to her/him.
- 55 The reviewer is very knowledgeable and knows the technical detail very well in the study by He et al. (2016). The reviewer is right that in He et al. (2016) study, "turnover times" were derived from decay parameters of a model and not by carbon stock divided by carbon flux. Our sentence "Up to 40% of soil C sequestration potential can be overestimated due to underestimation of C turnover time in current CMIP5 models (He et al., 2016)" reflects a main conclusion from that study. Besides, He et al. (2016) used the term "turnover time" in their paper. As Sierra et al. (2017) pointed out, there are many different ways to define
- 60 "turnover time" in the literature. We used the study by He et al. (2016) as one example to highlight the need to understand turnover time better instead of to define the term of turnover time at this stage. We were puzzled why the reviewer thought the paper by He et al. (2016) "is not suitable for motivating" our study.

Reviewer 1: A similar remark concerns the study of Friend et al. (2013) cited in the introduction: Friend et al. indeed calculate turnover time as carbon stock divided by carbon flux but as a diagnostic to test the validity of their simple carbon model, but

not for drawing any other conclusions from it that could be improved by using transit time. Hence, in my opinion the authors 65 handle the cited literature inappropriately to motivate their study and thereby give a wrong impression of their relevance.

Response: The reviewer used another sentence in the manuscript to question the motivation of our study. The sentence "Recent model inter-comparison study indicated that a major cause of uncertainty in predicting future terrestrial C sequestration is the

variation in C turnover time among the models (Friend et al., 2014)" was used to highlight the need to understand turnover

70 time better as well. We were very confused by reviewer's point that because Friend et al. (2014) did not mention transit time in their paper, we could not use it to motive our study on turnover time in relation with transit time. Overall, we are grateful to the reviewer for his/her critiques, which made us to carefully examine our manuscript again.

Reviewer 1: 2) The relevance of the results of the study is unclear

According to section 5 ('Conclusions') the study has two major results. The first (lines 357-361) concerns the development of
transit and turnover time in their simulations, in particular that they increasingly deviate from one another during the 21st century, and that the deviations are stronger in some regions than in others. But what to conclude from this? Is this a useful knowledge?

**Response:** We were not sure if the reviewer asked those two questions philosophically or practically. In practice, knowledge about the deviation between C transit time and turnover time in different regions under different scenarios (warming and [CO2]

- 80 rising) is useful for us to understand time characteristic of the ecosystem carbon dynamics. When we lump all pools and fluxes together to calculate turnover time by "stock over flux", the time characteristic is different from that of transit time when individual pools and fluxes are considered to be networked together to form compartmental dynamical system. Thus, our results provide information on how turnover time deviates from transit time in specific region and ecosystems. Because "stock over flux" is still the easiest way to measure how fast C cycle through ecosystem, it is not our purpose to persuade the
- 85 community to completely give up the method. Instead, we only provide information about how carbon transit time deviates with the turnover time in the non-steady state. We estimated both C transit time and turnover time globally. We assume the deviation between them is due to the lost information by lumping pools and fluxes to calculate the turnover time. The temporal and spatial estimates on the deviation tell us whether we can still use turnover time at a specific place and time.

If those are philosophical questions, we may not be able to answer either of the questions correctly. In this case, it is only 90 reviewer's perspective that matters.

**Anyway, in the revision, we added how this knowledge is practically useful in the Conclusion. (See Line 419-424)**

**Reviewer 1**: The second result (lines 363-368) is that transit time can be separated into contributions from MAC and ACC because the residual is small (see eq. (6) and Fig. 3). Therefore on first sight I indeed thought that this separation is an interesting idea that could help to understand better how transit time behaves under different forcings. The authors claim (lines

95 364/365) that MAC is determined by carbon input and ACC by "differential responses of various C pools to climate warming and rising atmospheric [CO2]". While this latter formulation is rather cryptic, I take from it that the authors think that one could pin down what affects MAC and ACC so that e.g. for scenarios of different CO2 and/or temperature rise one could understand why transit time would develop differently.

Response: We thank the reviewer for her/his carefully examining the separation of Mean Age Change (MAC) from Age

100 Composition Change (ACC) in our study. We agree that the contributions by MAC and ACC are not independent with each other, similarly as pool and flux influence each other. But the separation of pool and flux is so fundamental for any research

to understand carbon cycle. In this study MAC is related to the change caused by carbon age, which is equivalent to pool change in the pool-flux separation whereas ACC is more about the change caused by flux.

We understand that it may take time to fully comprehend new concepts about MAC and ACC. The separation was made according to derivative of an equation with two components. This method is very commonly used in many studies. For example, Koven et al. (2015) used this method to separate relative contributions of NPP vs. turnover time in influencing carbon sequestration. By separating the two terms, we can better understand mechanisms underlying the changes in transit time.

**Reviewer 1:** But I very much doubt that this separation helps understanding anything since there is no way to see how MAC and ACC are separately affected by the carbon inputs or the forcings: this is because MAC and ACC are not independent from

110 one another since they are both derived from the same development of carbon stored in the compartments (xi(t) in eq. (2) and eq. line 166). Hence a change of the carbon input into the system changes both MAC and ACC, and also a change in pool turnover time parameters by a changing climate (temperature, moisture) changes both MAC and ACC.

Response: The reviewer "very much doubt that this separation helps understanding anything since there is no way to see how MAC and ACC are separately affected by the carbon inputs or the forcings". We do not agree that contributions from different

115 reasons cannot be separated only because they are both partly affected by the same factor. There are many opposite examples in C cycle study. For examples, C pool and flux may both respond to change in C input. However, their responses have different ecological meanings.

Moreover, eq. (6) is the major equation to show the differences between MAC and ACC. MAC is mainly contributed by the change in age and ACC is mainly contributed by the change in composition.

120 Reviewer 1: Only if one could understand how climate and CO2 act differently on MAC and ACC this separation could contribute to a better understanding of transit time development in transient simulations.

Response: Reviewer was wondering "how climate and CO2 act differently on MAC and ACC". Our results have shown different responses of MAC and ACC to climate and CO2. Figure 3d shows that contribution from ACC (in blue) changes from negative to positive in response to C balance change, whereas MAC (in red) keep increasing. These features are very

125 similar to C pool and flux that C flux response faster than C pool to input change. More evidence that contributions from MAC and ACC vary spatially (Figure 4) also indicates the separation is helpful for our understanding.

Reviewer 1: That MAC and ACC have no individual meaning can also be seen directly in the simulation results: In the simulation with both forcings combined (simulation S3) the contribution to transit time from ACC is not even approximately the sum of the ACC contributions from the simulations with forcings separated (S1, S2), and the same is true for MAC. Hence,the behaviour of simulation S3 cannot be understood as combination of results from S1 and S2.

- 130 the behaviour of simulation S3 cannot be understood as combination of results from S1 and S2. Response: The reviewer also doubted that "MAC and ACC have no individual meaning" because the sum of individual effects (warming effect and [CO2] rising effect) does not equal to the combined effects. The non-additive MAC or ACC in response to warming and [CO2] rising is possibly due to the non-linear or interactive effects, which have commonly been found in other experimental and modelling studies. For examples, climate and rising atmospheric [CO2] affect GPP together, but usually their
- 135 co-effects are not equal to the sum of their individual effects (Norby and Luo, 2004; Luo et al., 2008; Leuzinger et al., 2011;

Campbell et al., 1997; Zhang et al., 2016). However, it does not necessarily mean climate effects and rising atmospheric [CO2] effects on GPP should not be separated.

Meanwhile, MAC and ACC do have individual meaning. In theory, Eqn (6) has illustrated that MAC represents the contribution from change in age and ACC represents the contribution from change in composition. In practice, results from

their responses (Figure 3d and Figure 4) have also confirmed they are completely different.In the revision, we added clarification in the Results. (See Line 245-248)

**Reviewer 1:** – In conclusion, I think this separation is only technical and pretty useless. In order to convince me from the opposite, the authors had to show me a case where it leads to an improved unterstanding.

Response: In practice, the contributions of MAC and ACC should be different among models. None of previous studies have

- 145 diagnosed those two contributions separately. Although total change in C transit time has been compared recently (Sierra, 2017), a thorough assessment on their individual contributions would provide more useful information. Otherwise, models are very likely to get the right answer with wrong reasons. By combining compartment models with different types of measurements via data assimilation techniques, we may be able to better constrain MAC and ACC respectively. Therefore, modelled C cycle can be better calibrated by constraining MAC and ACC against measurements in the future.
- 150 In the revision, we have added a paragraph in the discussion to show how the separation can be useful. (See Line 403-411) Reviewer 1: Concernig the other remarks related to the second result in lines 366-368, I think they are all wrong: (i) The calculation of turnover time by dividing stocks by fluxes is not assuming anything, it is simply a diagnostic that in the case of stationary states has a well defined meaning, but can, as a diagnostic, still be a useful concept (see my remark on the study by Friend et al. (2013) above).
- 155 Response: We appreciate that this reviewer clearly shows her/his view and perspective. He or she stand strongly for using turnover time. We agree that turnover time is a good and useful diagnostic, since both stock and fluxes are easy to measure. Our manuscript have admitted the advantages of C turnover time "C turnover time can be easily calculated from C stock over flux, both of which can be easily measured." (Line 379-380) However, would his/her statement "the calculation of turnover time is not assuming anything" be contradictory to "it is simply a diagnostic that in the case of stationary states has a well-
- 160 defined meaning"? Would "in the case of stationary states" be an assumption? But we hope the reviewer agrees that turnover time is calculated by lumping pools and fluxes together. As a scientist, he or she, we hope, will not be against research to explore other ideas related to time characteristics of carbon cycle.

**Reviewer 1**: (ii) Surely turnover time changes when MAC and ACC change, so that contrary to the authors claim it accounts for such changes.

165 Response: We mentioned "C turnover time does not account for changes in age structure and contribution fractions of different pools to ecosystem respiration." (Line 366-367). This sentence did not say "turnover time does not change". Obviously, C turnover changes when MAC and ACC changes. However, we care about whether the diagnostic really accounts for the certain critical information. The "change" in C turnover time with MAC and ACC is not really sufficient enough to accurately quantify the contributions from MAC and ACC to C cycle time characteristics.

170 Reviewer 1: Hence (iii) contrary to the claim by the authors one cannot conclude that transit time is a "better parameter". – A similar claim is found in the last scentence of the abstract where the conclusion is even weirder by saying that the use of turnover time instead of transient time may "lead to biases in estimating land C sequestration" – how could the mere calculation of a time scale affect the estimation of C sequestration?

Response: The reviewer seems to believe that there is no better diagnostic than turnover time. We partly agree but it should

- 175 depend on the specific case we are studying. Eg. Friend et al., (2014) identified that the source of the uncertainty in predicating land C sequestration C is mainly from turnover time. Qualitatively, turnover time and transit time behave similarly. Thus, the turnover time can be used in this case to point out a direction for model improvement. However, turnover time may not represent the time characteristic of carbon dynamics if we are interested in carbon sequestration in multiple pools, because turnover time uses lumped pools and fluxes for calculation.
- 180 Reviewer 1: 3) Some suggestions for improving the paper
  - There is one result of the paper surprisingly not mentioned in the conclusions that in my opinion makes an important contribution to land carbon research: This is the comparison of the CABLE results for transit time with the observational results by Carvalhais et al. (2014) in Fig. 2. When the Carvalhais et al. paper appeared, I thought it's nice that they produced a map of stocks divided by fluxes so that this turnover time can be used as a diagnostic to easily compare with results from
- 185 model simulations. But with the study under review here, we now know that despite the non-stationarity of todays carbon cycle, turnover times agree well with transit times (Fig. (5)) so that the observational turnover times of Carvalhais et al. (2014) can indeed be interpreted as proper carbon ages. And that the zonal distribution of CABLE results matches those of Carvalhais et al. quite well provides additional credit to this conclusion.

Response: Thanks for the great suggestion to improve our manuscript. We have included this point in the discussions to support

190 Carvalhais et al. (2014). (See Line 323-330)

**Reviewer 1:** Hence, what I propose is that you focus your study on the question to what extend the observational estimates of turnover time by Carvalhais et al. (2014) (and if possible also those by Bloom et al. (2016) Fig. 3) can be interpreted as proper ages. In this respect it also interesting to see that shortly in the future this doesn't work any more. For your paper this would mean that you drastically shorten it by dropping anything else (i.e. in particular the simulations S1, S2 and all stuff relating to

195 the separation of transit time into contributions from ACC and MAC). With such changes I think a resubmission could make sense.

**Response:** This reviewer clearly has her/his own preference and tried to fit our study into his/her perspective. Hopefully this review is not Dr. Carvalhais or his associates. We do not believe Dr. Carvalhais or his associates would be such self-serving. Nonetheless, these are very constructive suggestions. We fully understand the reviewer's point and do see some interesting

200 conclusions being drawn in this way. It is very interesting that the reviewer states "In this respect it also interesting to see that shortly in the future this doesn't work any more." We thought we just did what the reviewer suggested us to do. Our analysis indicates that turnover time works quite well now and until the middle of this century before it significantly deviates from transit time. To do what the reviewer suggested us to do, we need the full length of the manuscript. In the revision, we have

added a paragraph to discuss how the deviation could be used to address the question to what extend the turnover time by

205 Carvalhais et al. (2014) is able to be interpreted as proper time characteristics in C cycle. (See Line 323-330)

**Minor Remarks**

235

**Reviewer 1:** • For a resubmission a better polished text would be appreciated. The current text is mostly understandable but the English shows quite some deficits (an annoyingly plentitude of missing articles; wrong grammar (lines 41, 47, 127, 154,

210 156, 175, 177, 178, 261, 357, 479; Supplement scattered with errors); incomplete formulations (lines 56, 139, 167, 210); wrong or missing preposition (lines 99, 333), ununderstandable formulations (line 168, 196, 274)). And results should be presented in present tense not in past tense as e.g. in the abstract.

Response: We have carefully checked grammar, formulations, etc. and used present tense.

Reviewer 1: • I do not see why in addition to the terms 'transit time' and 'turnover time' one needs the equivalent use of

215 namings 'Olson method' and 'Rasmussen method', respectively, this is only confusing – by whatever method you compute turnover or transit time, they remain the same.

Response: Thanks for pointing out the confusion. In the revision, we will use only transit time and turnover time avoid the confusion.

Reviewer 1: • In the model description for CABLE you refer for the photosynthesis part to a paper by Farquhar that refers to

220 the C3 pathway only. What does it mean for the realism of your simulations that CABLE is not accounting for C4 photosynthesis, happening at huge areas wordwide?

**Response:** Sorry for the misleading, but CABLE does account for C4 photosynthesis, which follows Kowalczyk et al. (2006). In the revision, we have added this information to clarify. (See Line 101-102)

Reviewer 1: • What about land use change? This seems to be not accounted for in CABLE, but replacement of forests with

agricultural lands could in principle speed up the land carbon cycle by one magnitude (maybe forests: 30 years vs. agriculture 1 year).

**Response:** Land use change is not accounted for in CABLE and we have discussed the possible bias caused by this in the first paragraph in discussion section 4.3. (See Line 350-359)

Reviewer 1: • What about natural vegetation? How does it change in your CABLE simulations?

230 Response: CABLE is not a dynamics vegetation model, which means natural vegetation distribution is static. In the revision, we have added some discussions on how potential changes in natural vegetation distribution could influence the estimates of transit time and turnover time. (See Line 361-369)

**Reviewer 1:** • For the historical period CABLE was forced by CRU-NCEP data, while for 2006-2100 simulation data from CESM were used. How good do these simulated climate data fit at the transition period around 2005/2006 to the historical values, concerning e.g. the global and zonal levels of land temperature, precipitation, and radiation?

Response: To show the forcing in transition period, we have added the annual variation of 8 meteorological forcing variables and  $[CO_2]$  data from 1901 to 2100 in the Appendix B.

Reviewer 1: • When you introduce the Rasmussen method to calculate transit time it would be good to mention that this

- 240 approach works only for linear box models. Is CABLE really of this type? You could demonstrate this by listing in the appendix the box-model equations for CABLE (like in section 8 of the Rasmussen et al. paper) – this would also help to make precise what at all your mathematical symbols mean. – I wonder about the applicability of the Rasmussen et al. approach because I would think that e.g. the phenology introduces some non-linearity in the dynamics of leaf carbon since leaf area cannot grow beyond a certain value depending on vegetation type – but maybe CABLE works differently. And what about structural
- 245 allometries between different plant parts (stems, roots, leaves) that are also non-linear? Response: These questions and suggestions are very helpful to improve the manuscript. We have given more details about CABLE in appendix C. Yes, C cycle in CABLE, even with phenology processes, can be considered as a linear model. In the deciduous plant functional type, CABLE's phenology only changes the leaf turnover rate and allocation fraction in spring and fall. When LAI grows over the upper limit, CABLE will set the leaf C allocation to "0" in order to prevent further leaf growth.
- 250 The dynamics of both turnover rate and C input are determined by time-dependent environmental scalars. Because the environmental scalars are independent on C pool sizes in most cases, the model can be considered as a linear. In addition, CABLE does not include the structural allometries.
  Local determined by the environmental scalars are independent on C pool sizes in most cases, the model can be considered as a linear. In addition, CABLE does not include the structural allometries.

In the revision, we have clarified the required linear condition in Rasmussen method (See Line 152). Moreover, we have also listed equations for CABLE C cycle in the Appendix C.

255 Reviewer 1: • Fig. 1: (i) Title Fig. 1a: Transient→ Transit. (ii) Make the scale numbering for both plots of Fig. 1 better readable (e.g. in steps of 5 or 10 years, or, if logarithmic, use other round numbers, but definitely not something like 3623).

Response: We have revised as suggested. (See Line 575)

**Reviewer 1:** (iii) Are the colors at the edges of e.g. Antarctica and Greenland really a result of your simulations, or is it a plotting artefact e.g. from your grid cell interpolations?

260 Response: Those colors at the edges are the real results of the simulations. The red indicates that C transit time and mean age are really high in the high latitude region. In contrast, the edges of islands at lower latitudes, e.g., Hawaii, are relative low. Reviewer 1: • Lines 147-150: I guess that this paragraph should say that the authors solve equation (2) by an Euler method starting from zero land carbon – this should be stated more clearly. Response: We have followed the suggestion to describe more clearly. (See Line 160)

**5 D** (1, 2, 3) **When 1** (2, 3) **When 1** (1, 4) **When 1** (2, 3) **When 1**

265 Reviewer 1: • Fig. 2: Why don't you also plot turnover time from CABLE? This would make it even more clear that transient and turnover time match well for this period of time.

Response: Thanks for the great suggestion. We have added turnover time in Figure 2. (See Line 216-217 and 582-586) Reviewer 1: • Why do you talk of "permafrost areas" instead of e.g. "high latitudes"? I guess that CABLE is not accounting for permafrost.

270 Response: We agree that "high latitudes" is more accurate. We have revised the wording.

Reviewer 1: • Fig. 6g: You attribute the small difference in turnover and transit time for the stationary state to the presence of

the seasonal cycle that makes the system non-stationary. Can this explain the increase of this difference beyond 60°N?

**Response:** Yes, conceptually, the significant bias in high latitude should be due to the seasonal cycle. We have illustrated in the results. (See Line 269-270)

275

- Line 361~369: added "In contrast to the static vegetation distribution used in CABLE, natural vegetation distribution may change over time in the real world. C transit time and turnover time may further deviate under natural vegetation dynamics. However, whether forest will expand or dieback in a future warming world is still quite unknown.
  Previous studies variously conclude due to their focus on different areas with different methods (Masek, 2001; Soja et al., 2007; Cox et al., 2004; Cox et al., 2013). Nevertheless, most bioclimatic models consistently suggest temperate and boreal biomes rapidly increase in area under warming (Kirilenko and Solomon, 1998). If the forest species, which stores more C in slow-turnover tissue, takes over the grass species, which stores more C in fast-turnover tissue, the expansion of forest may increase C transit time significantly. However, C turnover time by lumping all different C compartments together may underestimate such changes."

Line 152: added "Note that this equation works only for linear models.".

- 50 Line 160: the sentence has been revised into "we obtain the steady state C ages in each compartment by solving Eqn (2) with an Euler method."
  - Line 216~217: added "Moreover, the simulated latitudinal pattern of C transit time almost overlaps with C turnover time, which also evident that C cycle is still near the steady state at present day."
- 55

60

65

- Line 269~270: added "because seasonal soil frozen-thaw processes in this region lead to the strong seasonal cycle of the soil decomposition and violate the steady state assumption of the C turnover time."
- Line 24~26: sentence has been revised: "Warming increases C turnover time by 2.4 years and transit time by 11.8 years in 2100 relative to that at steady state in 1901"
- Line 583~586: added "Our assumption, which is also used by some other ecological studies (Trumbore, 2000), is that present-day ecosystem C cycle is closed to the steady state. Especially, in 1980s and 1990s, global land C uptake from Global Carbon Project (GCP) is about 0.8 GtC yr-1 with an uncertainty of 0.6 GtC yr-1, which is not significant compared to current decade with global land C uptake is 2.7 GtC yr-1."

**Ecosystem carbon transit versus turnover times in response to climate warming and rising atmospheric CO2 concentration**

Xingjie Lu1,2,3, Ying-Ping Wang3, Yiqi Luo2,4 and Lifen Jiang2

1School of Atmospheric Sciences, Sun Yat-sen University, Guangzhou 510275, China
 2Center for Ecosystem Science and Society, Department of Biological Sciences, Northern Arizona University, Flagstaff 86011, USA
 3CSIRO Oceans and Atmosphere, Aspendale 3195, Australia
 4Department for Earth System Science, Tsinghua University, Beijing 100084, China

10

Correspondence to: Xingjie Lu (xingjie.lu@nau.edu)

- 15 Abstract. Ecosystem carbon (C) transit time is a critical diagnostic parameter to characterize land C sequestration. This parameter has different variants in the literatures, including a commonly used turnover time. However, we know little about whether whether -transit time or turnover time better is a better diagnostic parameter to represents carbon cycling through multiple compartmentpools under non steady state However, neither of them has been carefully examined under transient C dynamics in response to climate change. In this study, we estimated both C turnover time as defined by the conventional stock-
- 20 over-flux (i.e., Olson method) and mean C transit time as defined by the mean age of C mass leaving the system (i.e., Rasmussen method). We incorporated them into Community Atmosphere-Biosphere-Land Exchange model (CABLE) to estimate C turnover time and transit time, respectively, in response to climate warming and rising atmospheric [CO2]. Modeling analysis showsed that both C turnover time and transit time increased with climate warming but decreasesd with rising atmospheric [CO2]. Warming The-increases of C turnover time with respect to steady state with under warming was is
- 25 estimated to beby 2.4 years with Olson method whereasand the transit time increased by 11.8 years in 2100 relative to that at steady state in 1901 with Rasmussen method. During the same period, The decrease with rising atmospheric [CO2] decreases C turnover time by was is estimated to be 3.8 years with Olson methodin C turnover time and 5.5 years with Rasmussen method in transit time by 5.5 years. Our analysis based on Rasmussen method showsed that 65% of the increase in global mean C transit time with climate warming results from the depletion of fast-turnover C pool. The remaining 35% increase results
- 30 from accompanied changes in compartment C age structures. Similarly, the decrease in mean C transit time with rising atmospheric [CO2] results approximately equally from replenishment of C into fast-turnover C pool and subsequent decrease in compartment C age structure. Greatly different from the Rasmussen methodtransit time, the Olsen methodturnover time, which does not account for changes in either C age structure or composition of respired C, underestimated impacts of either warming or rising atmospheric [CO2] on C diagnostic time and potentially lead to biases in estimating land C sequestration in
- 35 multi-compartmental ecosystems.

**1** Introduction**

70

Terrestrial ecosystem plays an important role in mitigation of climate change through sequestering carbon (C) from the

- 40 atmosphere. Terrestrial C storage is co-determined by C input and C transit time, which is defined as the mean age of C mass leaving the system (Luo et al., 2001; Taylor and Lloyd, 1992; Nir and Lewis, 1975; Sierra et al., 2016; Manzoni et al., 2009; Eriksson, 1971; Bolin and Rodhe, 1973). As transit time cannot be easily estimated from observation, its variant, C turnover time, has been commonly used in the literature (Sierra et al., 2016). Recent model inter-comparison study indicated that a major cause of uncertainty in predicting future terrestrial C sequestration is the variation in C turnover time among the models
- 45 (Friend et al., 2014). Up to 40% of soil C sequestration potential can be overestimated due to underestimation of C turnover time in current CMIP5 models (He et al., 2016). The C turnover time has been mostly estimated with a conventional stock-over-flux method (Carvalhais et al., 2014; Chen et al., 2013; Yan et al., 2017), which is probably first introduced by (Olson, 1963) and - Hereafter in this paper, we call assume that "C turnover time" also indicates the use of "stock over flux" methodit Olson method. The Olson method C turnover time is based on a steady-state assumption. In response to climate change,
- 50 terrestrial ecosystem C dynamics move away from steady states to be at dynamic disequilibrium (Luo and Weng, 2011). Estimation of C turnover time with Olson method-likely deviates from C transit time in response to climate change (Sierra et al., 2016). It is not clear how much the do-estimates of C turnover time deviates from mean C transit time and what causes their deviations under climate change.
- 55 The C transit time as the mean age of C mass leaving the system can be estimated only from age structure of C atoms in a multi-compartment ecosystem. In contrast, the C turnover time is estimated without any information of age structure of C atoms among compartments. Thus, C turnover time is equivalent to C mean transit time only in the autonomous (i.e., time-invariant) linear-system at steady states (Sierra et al., 2016) with three-two conditions to be satisfied. The first condition is that C fluxes and turnover rates of each-individual pools do not change with time (i.e., time invariant or autonomous). The second
- 60 is that C turnover rate of each pool is not a function of pool size as in linear C transfer models. The third second is that C influx to each pool equals to C efflux from the pool (i.e., at steady state). However, the autonomous, linear and steady state system conditionsat steady state is are a very usually too strict to completely meete ondition for real-world ecosystems. For examples, Ecosystem cosystem C input via photosynthesis has diurnal variation, seasonal cycle, and inter-annual variability. C turnover time also exhibits strong seasonal variation (Luo et al., 2017). With seasonal cycles and inter-annual variability in both C input
- and turnover time, ecosystem C cycle is rarely at steady state rather than mostly at dynamic disequilibrium (Luo and Weng 2011). Therefore, C turnover time hardly equals C transit time in the real world, especially when land C cycle is under transient dynamics in response to climate change.

The estimates of C transit time requires information of C age structure in ecosystems so that the mean age of the C atoms at the a time when they leave the system can be calculated (Manzoni et al., 2009). In a multi-compartmental ecosystem, the C

age within each compartment is represented by a single compartment C mean age and different compartments have different C mean ages (Rasmussen et al., 2016). Thus, the C transit time is the weighed mean of ages of C atoms leaving different compartments according to the contributing fraction of C loss from each pool to the total C loss. Hereafter in this paper the term "C transit time" This will also indicate this calculation of transit time hereafter is called from (Rasmussen et al., 2016).

[revised manuscript text omitted]
 CO2, for example, If more young-age C enters influx into a compartment is-more than the C effluxit leaves. aseg. under elevated CO2, C age structure in the compartment will become becomes younger (i.e., young-age C replenishment). Subsequently, ecosystem mean C transit time will reduce. The third term refers to residuals that cannot be explained by the previous two terms.

**3. Results**

**195 3.1 Global steady-state patterns of ecosystem C transit time**

The global ecosystem C transit time at steady state estimated by Rasmussen method-generally shows a latitudinal variation pattern (Fig. 1). The high values (greater than 70 years) are simulated not only in high latitude regions, such as northern Russia, northern Europe, and northern Canada but also in high altitude regions such as Tibet plateau. Small values in C transit time (less than 30 years) are simulated in tropical rainforest, such as Amazon forest, Conga forest, and Indonesia forest. Ecosystem

200 C transit times in some grass lands in middle-south Africa, south America, Southern Great Plains of US, and central north Australia (savanna) sometime are even smaller than that in tropical forest. The spatial patterns of the ecosystem C mean age are quite similar with the patterns of C transit time. However, the magnitude is significantly higher than ecosystem C transit time. The ecosystem C mean age ranges from 118 years to 7952 years, whereas ecosystem C transit time ranges only from 13 years to 341 years.

**205**

The global latitudinal pattern of C transit time estimated from Rasmussen method-in 1982-2005 is consistent with the observation-based pattern of turnover time (Fig. 2). The latter is estimated at each grid cell globally by Olson-"stock-over-flux" method to divide ecosystem C storage by gross primary productivity (GPP) (Carvalhais et al., 2014). The magnitude of the estimate is mostly within the uncertainty range of the observation-based pattern. We compared estimated C transit time in 1982-2005 with the turnover time, partly to match modelled values with contemporary observations, which is based on and partly due to the fact that terrestrial C cycle is still approximately at a quasi-steady state between 1982 and 2005. Over the 1980s and 1990s, the annual average of global net land carbon sink estimated from Global Carbon Project (GCP) is about 0.8 GtC yr-1 with an uncertainty of 0.6 GtC yr-1. As a reference, the annual average of net land carbon sink in recent decade (2007-2016) is 2.3 GtC yr-1 with an uncertainty of 0.7 GtC yr-1 (Le Quere et al., 2018). The net change of global land carbon in 1980s and 1990s is not that significant, which indicates land C cycle has not moved away too far from the steady state. Moreover, the simulated latitudinal pattern of C transit time almost overlaps with C turnover time, which also evident that C cycle is still near the steady state at present -day. Annual C turnover time using Olson method-theoretically equals to C transit time from Rasmussen method when C cycle is close to the steady state (Sierra et al., 2016).

**3.2 Responses of global C mean transit time to climate change**

In 200-year simulation, global ecosystem C mean transit time increasesd by 11.8 years in response to climate warming (S1) and decrease by 5.6 years in response to rising atmospheric [CO2] (S2) (Fig. 3a). When climate warming and rising atmospheric [CO2] forced together (S3), C transit time decreasesd by 1.6 years. The increase in C transit time in S1 is not significant in the 20th century but substantial in the 21st century. Oppositely, the decrease in C transit time in S2 is steady before 2060 but slow down afterward. Mean C transit time in S3 decreases but with a smaller magnitude than that for S2 in the 21st century.

225

Across all the three scenarios, the most-majority (over 93.4%) of the changes in C transit time can be explained by the two combined changes in compartment C age structure and respired C composition. Changes in the compartment C age structure and the respired C composition both significantly contributed to the total change in global C transit time. However, the contribution fraction varied-vary among the three scenarios at different time. In climate warming scenario (S1), respired C composition changes contribute about 70% of the increase in C transit time in the 21st century (Fig. 3b). In the rising atmospheric [CO2] scenario (S2), respired C composition change and C age structure change contribute equally (Fig. 3c). When coupling climate warming and rising atmospheric [CO2] together in S3, respired C composition change significantly contributes only in the middle of 200-year simulation (around year 2000), but little at the end of the 21st century. The contribution of C age structure change to the change in C transit time gradually increases.

- 235
- The increase in C transit time in climate warming scenario (S1) is the most significant from low latitude regions in southSouth America and Africa (Fig. 4a). Respired C composition change explains most of these regional changes (Fig. 4c). The decrease in C transit time in rising atmospheric [CO2] scenario (S2) is evenly simulated all over the world (Fig. 4d). Respired C composition change also plays an important role in most regions except for northNorth Africa with little vegetation coverage.
  The C transit time in combined climate warming and rising atmospheric [CO2] scenario (S3) mostly decrease in northern hemisphere, but increase in some tropical grassland regions in South America and Africa (Fig. 4g). In those regions where C transit time decrease, compartment C age structure change due to fresh C replenishment explain most of the change in C transit time.
- 245 It should be nNoted that the response under combined effects (S3) is not a sum of thoseat from individual effects (S1 plus S2). The non-additive response to climate warming and rising atmospheric [CO2] is probably due to non-linear-interactive effects their interactions, which have been commonly found and widely studied-in many other-ecological researchstudies (
[revised manuscript text omitted]

uncertainty Especially, uncertainties in C cycle time characteristics are the major reason to hinder precise future prediction in land C sequestration (Friend et al., 2014; He et al., 2016). In the transient state, the response in C cycle time characteristics are usually extraordinarily complicated. Since thisOur study has provided insights shown that to separate the change in C transit time can be separated into two components, C composition change and C age change. a more thorough model a Assessment on the twoindividual components would provide additional critical model constraints on model projections. To further
constrain C transit time inthrough its two components with observation, modelled C cycle and land C sequestration can be significantly improved based on this separation method.

**5. Conclusions**

- This study explores how global ecosystem C transit time deviates with-from the turnover time under climate warming and 415 rising atmospheric [CO2]. Although both global ecosystem C transit time and turnover time increase in response to climate warming and decrease in response to rising atmospheric [CO2], their deviations increase with time in all the three climate change scenarios. In 2100, the deviations are high in tropical regions under climate warming scenario (S1) and rising atmospheric [CO2] scenario (S2), and in permafrost-high latitude regions under S1 and combined change scenario (S3). Knowledge about the deviation between C transit time and turnover time in different regions under different scenarios
- 420 (warming and [CO2] rising) is useful for us to understand time characteristic of the ecosystem carbon dynamics. When we

lump all pools and fluxes together to calculate turnover time by "stock over flux", the time characteristic is different from that of transit time when individual pools and fluxes are considered to bewithin a networked together to form-compartmental dynamical-system. Thus, in practice, our results from S3-provide information on how future-turnover time in the future could deviates from transit time in specific regions and natural ecosystems under different climate change scenarios. In addition, our

425 results from S1 and S2 are special cases that help us identify what climate change factors are critically contributed to the biases in specific regions, so that we gain further insights on the cause of the biases.

[revised manuscript text omitted]

- He, Y. J., Trumbore, S. E., Torn, M. S., Harden, J. W., Vaughn, L. J. S., Allison, S. D., and Randerson, J. T.: Radiocarbon constraints imply reduced carbon uptake by soils during the 21st century, Science, 353, 1419-1424, 10.1126/science.aad4273, 2016.
  Huang, Y. Y., Lu, X. J., Shi, Z., Lawrence, D., Koven, C. D., Xia, J. Y., Du, Z. G., Kluzek, E., and Luo, Y. Q.: Matrix approach to land carbon cycle modeling: A case study with the Community Land Model, Global Change Biol, 24, 1394-1404, 2018.
  Hurrell, J. W., Holland, M. M., Gent, P. R., Ghan, S., Kay, J. E., Kushner, P. J., Lamarque, J. F., Large, W. G., Lawrence, D., Lindsay, K.,
- 475 Lipscomb, W. H., Long, M. C., Mahowald, N., Marsh, D. R., Neale, R. B., Rasch, P., Vavrus, S., Vertenstein, M., Bader, D., Collins, W. D., Hack, J. J., Kiehl, J., and Marshall, S.: The Community Earth System Model A Framework for Collaborative Research, B Am Meteorol Soc, 94, 1339-1360, 10.1175/Bams-D-12-00121.1, 2013. Kelly, R. H., Parton, W. J., Hartman, M. D., Stretch, L. K., Ojima, D. S., and Schimel, D. S.: Intra-annual and interannual variability of ecosystem processes in shortgrass steppe, J Geophys Res-Atmos, 105, 20093-20100, Doi 10.1029/2000jd900259, 2000.
- 480 Kirilenko, A. P., and Solomon, A. M.: Modeling dynamic vegetation response to rapid climate change using bioclimatic classification, Climatic Change, 38, 15-49, 1998.

Koven, C. D., Riley, W. J., Subin, Z. M., Tang, J. Y., Torn, M. S., Collins, W. D., Bonan, G. B., Lawrence, D. M., and Swenson, S. C.: The effect of vertically resolved soil biogeochemistry and alternate soil C and N models on C dynamics of CLM4, Biogeosciences, 10, 7109-7131, 10.5194/bg-10-7109-2013, 2013.

485 Koven, C. D., Chambers, J. Q., Georgiou, K., Knox, R., Negron-Juarez, R., Riley, W. J., Arora, V. K., Brovkin, V., Friedlingstein, P., and Jones, C. D.: Controls on terrestrial carbon feedbacks by productivity versus turnover in the CMIP5 Earth System Models, Biogeosciences, 12, 5211-5228, 2015.

[revised manuscript text omitted]

\*\* Pre-industrial CO2 concentration is from CMIP5 dataset for the year 1901.

---

## Referee Report (RR1)

September 21, 2018

**General comments**

I very much appreciate that the authors took intense action to respond to all of my remarks and questions. Overall I see an improvement of the paper, although I still think that the separation of transit time into the components MAC and ACC (see below) still found in the revised paper is only of academic interest so that I do not see why to publish this. Nevertheless, its technically correct so that upon publishing one could leave it to the scientific community to follow my opinion or not. Another thing concerns the appraisal of 'transit time' over 'transient time' also in the revised paper. I can understand that the authors burn for their subject, but in my opinion they overshoot, so that I suggest that they revise the respective parts of the paper (see my more detailed comments below). A final thing is the multitude of language errors also present in the revised paper.

**Detailed comments**

In my review I had three major remarks. In the following I comment separately on the author's answers to them.

**Remark 1) from my review: The study is not well motivated**
From the response of the authors and also the remarks from reviewer #2, I understand now that by referring in the introduction to the papers of Friend et al. (2014) and He et al. (2016) the authors want to point out that much of the uncertainty for a realistic simulation of the land carbon cycle arises from the fact that the internal time scales are not well known. Unfortunately, they use here the term 'turnover time' that has a very specific meaning in their paper instead of e.g. a more neutral term like 'time scales', 'memory' or so. Thereby the introduction can be misunderstood as if the authors would like to justify their research by claiming that in the literature (particularly in those two papers) turnover time is used instead of – in the authors opinion more appropriate – transit time. This invited for the critique I expressed in my review to use those two papers for a justification of their study. So I suggest the authors revise their introduction to prevent such a misunderstanding. Thereby it will hopefully also get clearer that the main argument for their study may be found when following the hint from the end of the first paragraph in the introduction, namely that under transient conditions transit

time is different from turnover time – and pointing this out this may result in a proper justification for their study when it is added that only transit time has under these transient conditions a proper inner-theoretical meaning in the context of compartemental models.

In this connection I want to point out that I do not see any reason to qualify the use of turnover time against transit time – both have their advantages and disadvantages, as the authors well know (see their answer to my review). Therefore I find it inappropriate to call the difference of turnover time to transit time a 'bias' – it is simply a difference. And I also cannot follow the authors claim that when using transit time instead of turnover time one would solve the problem of the not well known internal time scales of the carbon cycle as e.g. expressed in the sentence (lines 376/377): "Estimating C transit times in the real world can help constrain projections in land C sequestration by C cycle models because C turnover time is a major source of model uncertainty (Friend et al., 2014; He et al., 2016)." Here the authors play with false cards, since they leave the impression that the use of turnover time instead of transient time is the problem that makes our knowledge of the carbon cycle so uncertain. But it is no question that Friend et al. (2014) and He et al. (2016) would have come to the same conclusions independently of using transit or turnover time – the problem is our incomplete knowledge of the internal time scales of the carbon cycle, not the use of transient time instead of turnover time. Moreover, land C cycle models could be well constrained with a good knowledge of either transient time or turnover time so that in this respect none of them has an advantage. The only advantage of transient time is that it has a proper inner-theoretical meaning even for transient states, but I do not see how this could lead to a practical advantage, except maybe in connection with the processing of labeled carbon ($^{13}C$, $^{14}C$) in vegetation. Therefore I suggest that the authors rethink their advocation of transient time thoughout the paper and revise the respective parts.

In addition to this, with the new second sentence in the abstract "However, we know little about whether transit time or turnover time better represents carbon cycling through multiple compartments under non steady state." the authors let the reader expect that the paper would answer the question which of the two times would 'better' represent the carbon cycle under these transient conditions – but this is not a question answered in the paper nor do I expect that it could be answered.

**Remark 2) from my review: The relevance of the results of the study is unclear**
I very much appreciate that in the resubmited paper it is now much clearer that a major result of this study is the matching of turnover and transit time during the historical period and their divergence in the future – and this should also clearly be expressed in the abstract. But concerning the relevance of the separation of transit time into MAC (mean age change) and ACC (age composition change) I have the feeling that we live on different planets so that I don't think that we could come to an agreement. None of the author's comments to my claim that this separation is useless convinces me of the opposite. In particular I still do not see where this separation could lead to an improved understanding. Nevertheless, in answering to this remark, the authors now indicate where

this separation may be useful (this is not 'understanding' but already something). In the revised paper they now write in lines 409-411 that by constraining "transit time through its two components [from] observations, modeled C cycle and land C sequestration can be significantly improved". I completely agree that if one could separate MAC and ACC one would have another diagnostic for comparison with models. But I very much doubt that ever these components could be measured because in view of the continuum of time scales in the land carbon cycle the discrete pools of models have no proper counterpart in our environment. It would be great if the authors could explain in their paper how to measure the two components in order to justify their claim that their method has the potential for 'significant' improvement.

**Remark 3) from my review: Some suggestions for improving the paper**
The authors have not taken up my suggestion to drop all material related to the separation of transit time into MAC and ACC, and I don't expect any agreement on this point (see previous comment). But I appreciate that most of my other suggestions have been taken up.

**Minor remarks**

- For completeness it would be good to explain that in the formulas (1) to (5) the number of compartments $d$ depends on the application: In the calculation for a single grid cell it is the number of pools from all vegetation types in that grid cell, while for global numbers it is the total number of all carbon pools in all grid cells worldwide – this would make clear that e.g. the transit time shown in Fig. 3a is not a global average of Fig. 1a (for the particular years).
- Line 81: What are 'contributing fractions'?
- Line 174: What is "age-mass C"?
- After now dropping the two terms 'Olson method' and 'Rasmussen method' in the revised paper, the authors may want to use different indices than 'o' and 'r' to distinguish turnover and transit time.
- The caption of Fig. 2 has been expanded, explaining now that for the current dacade land carbon uptake is much larger than for previous decades – I don't understand the reason for this explanation. And why is it your 'assumption' that today's C cycle is close to equilibrium – all your paper is about the transient state.
- I find the caption of Fig. 6 hard to follow. Maybe one could help the reader by adding names to the rows and columns of the figure by e.g naming the first row 'transit time', the second row 'turnover time', the third row 'comparison transit/turnover time', and first column 'steady state', second column '$\Delta\tau/\tau_{1900s}$', and third column '$\tau_{2090s} - \tau_{1900s}$'.

---

## Author Response (AR2)

Dear editor and reviewer:

We have responded point-by-point to reviewer's comment and revised the manuscript accordingly. In the revision, we mainly explained the details on how MAC (mean age change) and ACC (age composition change) in the real-world ecosystem can be measured. We hope you find the response and revision satisfactory.

We listed reviewer's comments and our point-by-point responses in **blue**.

Regards,

Xingjie Lu On behalf of all co-authors

**General comments**

I very much appreciate that the authors took intense action to respond to all of my remarks and questions. Overall I see an improvement of the paper, although I still think that the separation of transit time into the components MAC and ACC (see below) still found in the revised paper is only of academic interest so that I do not see why to publish this. Nevertheless, its

15 technically correct so that upon publishing one could leave it to the scientific community to follow my opinion or not.

Response: We appreciate reviewer's open-minded attitude. We hope that the separation of transit time into MAC and ACC is not only technically correct, but also practically useful. In the revision, we added examples on using MAC and ACC to constrain model from observation. (See Line 394-399)

Another thing concerns the appraisal of 'transit time' over 'transient time' also in the revised paper. I can understand that the

20 authors burn for their subject, but in my opinion they overshoot, so that I suggest that they revise the respective parts of the paper (see my more detailed comments below).

Response: We thank reviewer's careful consideration on how we should appraise new concept 'transit time' with existing concept 'turnover time'. Each point from reviewer has been deliberated to ensure the wording for both transit time and turnover time is appropriate.

25 A final thing is the multitude of language errors also present in the revised paper.

Response: Thanks for pointing out. We have carefully check over the language errors again through the paper.

Detailed comments

In my review I had three major remarks. In the following I comment separately on the author's answers to them.

Remark 1) from my review: The study is not well motivated

From the response of the authors and also the remarks from reviewer #2, I understand now that by referring in the introduction to the papers of Friend et al. (2014) and He et al. (2016) the authors want to point out that much of the uncertainty for a realistic simulation of the land carbon cycle arises from the fact that the internal time scales are not well known. Unfortunately, they use here the term 'turnover time' that has a very specific meaning in their paper instead of e.g. a more neutral term like 'time scales', 'memory' or so. Thereby the introduction can be misunderstood as if the authors would like to justify their research by claiming that in the literature (particularly in those two papers) turnover time is used instead of - in the authors opinion more appropriate - transit time. This invited for the critique I expressed in my review to use those two papers for a justification of their study. So I suggest the authors revise their introduction to prevent such a misunderstanding. Thereby it will hopefully also get clearer that the main argument for their study may be found when following the hint from the end of the first paragraph in the introduction, namely that under transient conditions transit time is different from turnover time - and pointing this out this may result in a proper justification for their study when it is added that only transit time has under these transient conditions a proper inner-theoretical meaning in the context of compartemental models.

Response: We appreciated that the reviewer disclosed his/her ideas, which help us make the paper clearer. In the revision, we added 'These examples highlight the importance of C turnover time in understanding C cycle uncertainties' to prevent misunderstanding. (See Line 42-43)

In this connection I want to point out that I do not see any reason to qualify the use of turnover time against transit time - both have their advantages and disadvantages, as the authors well know (see their answer to my review). Therefore I find it inappropriate to call the difference of turnover time to transit time a 'bias' - it is simply a difference.

Response: We do agree both C transit time and turnover time have their own advantages and disadvantages. As discussed in previous version (See Line 375-377), the advantage of C turnover time is that it can be easily measured, while the advantage of C transit time is that it is more theoretically correct under non-steady states. However, the whole paper is not to make a

comprehensive judgement on which one is better, but only focuses on their performance/deviations under non-steady state. In the revision, we replaced the word "bias" by "deviation" to avoid any judgmental implication. We have also clarified our focus and the importance to study the deviation. See Line 72-73

And I also cannot follow the authors claim that when using transit time instead of turnover time one would solve the problem of the not well known internal time scales of the carbon cycle as e.g. expressed in the sentence (lines 376/377): "Estimating C transit times in the real world can help constrain projections in land C sequestration by C cycle models because C turnover time is a major source of model uncertainty (Friend et al., 2014; He et al., 2016)." Here the authors play with false cards, since they leave the impression that the use of turnover time instead of transient time is the problem that makes our knowledge of the carbon cycle so uncertain. But it is no question that Friend et al. (2014) and He et al. (2016) would have come to the same conclusions independently of using transit or turnover time - the problem is our incomplete knowledge of the internal time scales of the carbon cycle, not the use of transient time instead of turnover time.

Response: We agree that the incomplete knowledge of internal time scale of carbon cycle instead of different concepts is the key issue for model improvement in C cycle projection. In the revision, we replaced the term "C turnover time" by "internal time scales of the carbon cycle" (See Line 392), and we described how we may improve our knowledge on the internal time scales of carbon cycle from more detailed measurements. (See Line 394-399)

Moreover, land C cycle models could be well constrained with a good knowledge of either transient time or turnover time so that in this respect none of them has an advantage. The only advantage of transient time is that it has a proper inner-theoretical meaning even for transient states, but I do not see how this could lead to a practical advantage, except maybe in connection with the processing of labeled carbon (13C, 14C) in vegetation. Therefore I suggest that the authors rethink their advocation of transient time thoughout the paper and revise the respective parts.

Response: We agree with the reviewer that land C cycle models can be better constrained with a good knowledge of either transient time or turnover time. Linking C transit time to isotope technique is a good direction for further studies. We have proposed these ideas in the manuscript. (See Line 383-389)

However, our manuscript theoretically explored how transient and turnover times deviate under non-steady state. In the revision, we try to point out their respective advantages and disadvantages.

In addition to this, with the new second sentence in the abstract "However, we know little about whether transit time or turnover time better represents carbon cycling through multiple compartments under non steady state." the authors let the reader expect that the paper would answer the question which of the two times would 'better' represent the carbon cycle under these transient conditions - but this is not a question answered in the paper nor do I expect that it could be answered.

Response: Thank the reviewer for carefully checking our sentences. We carefully read the sentence again "However, we know little about whether transit time or turnover time better represents carbon cycling through multiple compartments under non steady state." In the revision, we revised this sentence to be more accurate and avoid judgmental implication. (See Line 17)

Remark 2) from my review: The relevance of the results of the study is unclear

I very much appreciate that in the resubmited paper it is now much clearer that a major result of this study is the matching of turnover and transit time during the historical period and their divergence in the future - and this should also clearly be expressed in the abstract.

Response: Thanks for reviewer's understanding.

But concerning the relevance of the separation of transit time into MAC (mean age change) and ACC (age composition change) I have the feeling that we live on different planets so that I don't think that we could come to an agreement. None of the author's comments to my claim that this separation is useless convinces me of the opposite. In particular I still do not see where this separation could lead to an improved understanding. Nevertheless, in answering to this remark, the authors now indicate where this separation may be useful (this is not 'understanding' but already something). In the revised paper they now write in lines 409-411 that by constraining "transit time through its two components [from] observations, modeled C cycle and land C sequestration can be significantly improved". I completely agree that if one could separate MAC and ACC one would have another diagnostic for comparison with models. But I very much doubt that ever these components could be measured because in view of the continuum of time scales in the land carbon cycle the discrete pools of models have no proper counterpart in our environment. It would be great if the authors could explain in their paper how to measure the two components in order to justify their claim that their method has the potential for 'significant' improvement.

Response: Thanks reviewer for agreeing that the separation into MAC and ACC could provide new diagnostics for model comparison. Per request by the reviewer, we revised the manuscript (See Line 394-399) to explain measurements available to

those components in our environment. Many of the ecosystem pools, such as leaf C, wood C, root C pool, litter C pools, and soil C, can be measured separately. They provide plenty of heterogeneity information of ecosystem C pools. Isotope data from each component also can indicates the compartment mean age. Although discrete soil C pools may not be easy to separate, many datasets from field and laboratory measurements have been used to constrain multi-pool soil carbon models by using data assimilation technique (Xu et al., 2006; Liang et al., 2018).

Remark 3) from my review: Some suggestions for improving the paper

The authors have not taken up my suggestion to drop all material related to the separation of transit time into MAC and ACC, and I don't expect any agreement on this point (see previous comment). But I appreciate that most of my other suggestions have been taken up.

Response: Thanks.

Minor remarks

For completeness it would be good to explain that in the formulas (1) to (5) the number of compartments d depends on the application: In the calculation for a single grid cell it is the number of pools from all vegetation types in that grid cell, while for global numbers it is the total number of all carbon pools in all grid cells worldwide - this would make clear that e.g. the transit time shown in Fig. 3a is not a global average of Fig. 1a (for the particular years).

Response: Thanks for carefully studying our methods. It is true that the compartments also represent differently in single grid cell and in global scale. We have added clarification in the revision (See Line 185-190)

Line 81: What are 'contributing fractions'?

Response: We have replaced 'contributing fraction' with only 'fraction'. Thanks for carefully reading.

Line 174: What is "age-mass C"?

Response: We have replaced 'respired age-mass C' with 'products of respired C mass and C age'

After now dropping the two terms 'Olson method' and 'Rasmussen method' in the revised paper, the authors may want to use different indices than 'o' and 'r' to distinguish turnover and transit time.

Response: We have replaced the indices 'o' and 'r' with 'to' and 'ts' to note 'turnover' and 'transit' respectively. Thanks for carefully reading.

The caption of Fig. 2 has been expanded, explaining now that for the current dacade land carbon uptake is much larger than for previous decades - I don't understand the reason for this explanation. And why is it your 'assumption' that today's C cycle is close to equilibrium - all your paper is about the transient state.

Response: Thanks for pointing out this unclear part. One purpose to compare C transit time with observed C turnover time is to validate the modeled C transit time, which is always important for modeling study. However, C transit time theoretically equals to turnover time only at steady state. To check the steady state of global C cycle during the data period (1982 to 2005) is critical. Decadal land C balance reflects whether ecosystem C is in a steady state or non-steady state. Small number of mean C balance (0.8 GtC yr$^{-1}$) prove the quasi-steady state in 1980s and 1990s. Meanwhile, the increasing C sink in current decade also shows how system is gradually driven from steady state to non-steady state.

Yes, all the paper is focused on the transient state. However, we also mentioned "In response to climate change, terrestrial ecosystem C dynamics move away from steady states to dynamic disequilibrium" in the introduction. From 1901 to 2100, the transient state starts with a quasi-steady state and then move to the non-steady state, which also explain why C transit time is not significantly different from C turnover time until 2050. In the revision, we have clarified these points and remove the comparison with current decade to avoid misleading. (See Line 569-574)

I find the caption of Fig. 6 hard to follow. Maybe one could help the reader by adding names to the rows and columns of the figure by e.g naming the first row 'transit time', the second row 'turnover time', the third row 'comparison transit/turnover time', and first column 'steady state', second column '$\Delta\tau/\tau_{1900s}$', and third column '$\tau_{2090s}-\tau_{1900s}$'.

Response: Thanks for the great suggestions. We have added name for each row and column. (See Line 597)

Liang, J., Xia, J., Shi, Z., Jiang, L., Ma, S., Lu, X., Mauritz, M., Natali, S. M., Pegoraro, E., and Penton, C. R.: Biotic responses buffer warming-induced soil organic carbon loss in Arctic tundra, Global Change Biol, 2018.
Xu, T., White, L., Hui, D. F., and Luo, Y. Q.: Probabilistic inversion of a terrestrial ecosystem model: Analysis of uncertainty in parameter estimation and model prediction, Global Biogeochem Cy, 20, Artn Gb2007, Doi 10.1029/2005gb002468, 2006.

**List of major relevant changes:**

Line 17: Sentence has been revised: "However, we know little about how different are transit time and turnover time in representing carbon cycling through multiple compartments under non steady state"

Line 30, 79, 245, 258, 262, 264, 268, 281, 283, 289, 308, 339, 592, 595, 600-602, 605: "bias" has been replaced with "deviation"

Line 42-43: Sentence has been added: "These examples highlight the importance of C turnover time in understanding C cycle uncertainties".

Line 66, 70, 72: "contributing fraction" has been replaced with "fraction".

Line 73-74: Sentence has been revised: "It is necessary to understand the theoretical deviation between C transit time and C turnover time under non-steady state."

Line 155, 156, 166, 176, 379, 389, 593, 594, 596, 598-600, 602: replace "$\tau_R$" with "$\tau_{ts}$" and replace "$\tau_o$" with "$\tau_{to}$"

Line 164: replace "respired age-mass C" with "products of respired C mass and C age".

Line 185-190: Add a paragraph to specifically describe the calculation of average on three different spatial scale.

Line 392: "C turnover time" has been replaced with "internal time scales of the carbon cycle"

Line 394-399: Details on how to measure composition change and age change have been added.

Line 569-574: Explanation on the Fig. 2 caption has been added.

Line 587: Names have been added for rows and columns in Fig. 6.

**Ecosystem carbon transit versus turnover times in response to climate warming and rising atmospheric CO₂ concentration**

Xingjie Lu[1,2,3], Ying-Ping Wang[3], Yiqi Luo[2,4] and Lifen Jiang[2]

[1]School of Atmospheric Sciences, Sun Yat-sen University, Guangzhou 510275, China
[2]Center for Ecosystem Science and Society, Department of Biological Sciences, Northern Arizona University, Flagstaff 86011, USA
[3]CSIRO Oceans and Atmosphere, Aspendale 3195, Australia
[4]Department for Earth System Science, Tsinghua University, Beijing 100084, China

*Correspondence to*: Xingjie Lu (xngj.lu@gmail.com)

**Abstract.** Ecosystem carbon (C) transit time is a critical diagnostic parameter to characterize land C sequestration. This parameter has different variants in the literature, including a commonly used turnover time. However, we know little about  how different are transit time  and turnover time in representing carbon cycling through multiple compartments under non steady state. In this study, we estimate both C turnover time as defined by the conventional stock-over-flux and mean C transit time as defined by the mean age of C mass leaving the system. We incorporate them into Community Atmosphere-Biosphere-Land Exchange model (CABLE) to estimate C turnover time and transit time, respectively, in response to climate warming and rising atmospheric [$CO_2$]. Modeling analysis shows that both C turnover time and transit time increase with climate warming but decrease with rising atmospheric [$CO_2$]. Warming increases C turnover time by 2.4 years and transit time by 11.8 years in 2100 relative to that at steady state in 1901. During the same period, rising atmospheric [$CO_2$] decreases C turnover time by 3.8 years and transit time by 5.5 years. Our analysis shows that 65% of the increase in global mean C transit time with climate warming results from the depletion of fast-turnover C pool. The remaining 35% increase results from accompanied changes in compartment C age structures. Similarly, the decrease in mean C transit time with rising atmospheric [$CO_2$] results approximately equally from replenishment of C into fast-turnover C pool and subsequent decrease in compartment C age structure. Greatly different from the transit time, the turnover time, which does not account for changes in either C age structure or composition of respired C, underestimated impacts of either warming or rising atmospheric [$CO_2$] on C diagnostic time and potentially lead to deviations in estimating land C sequestration in multi-compartmental ecosystems.

**1 Introduction**

35  Terrestrial ecosystem plays an important role in mitigation of climate change through sequestering carbon (C) from the atmosphere. Terrestrial C storage is co-determined by C input and C transit time, which is defined as the mean age of C mass leaving the system (Luo et al., 2001; Taylor and Lloyd, 1992; Nir and Lewis, 1975; Sierra et al., 2016; Manzoni et al., 2009; Eriksson, 1971; Bolin and Rodhe, 1973). As transit time cannot be easily estimated from observation, its variant, C turnover time, has been commonly used in the literature (Sierra et al., 2016). Recent model inter-comparison study indicated that a

40  major cause of uncertainty in predicting future terrestrial C sequestration is the variation in C turnover time among the models (Friend et al., 2014). Up to 40% of soil C sequestration potential can be overestimated due to underestimation of C turnover time in current CMIP5 models (He et al., 2016). These examples highlight the importance of C turnover time in understanding C cycle uncertainties. However, The 
[revised manuscript text omitted]

185 In this study, the C age dynamics and diagnostics represented by Eqn (1)-(6) are implemented into CABLE model global simulations. In the synthesisDuring our analysis of results, C age and transit time are averaged at different spatial scales, i.e. grid cell scale, latitudinal scale, and global scale. Pool sizes ($x_i$), pool ages ($a_i$) and pool-to-pool fluxes ($F_{ij}$) in Eqn (1)-(6) uses the grid cell mean, latitudinal mean and global mean of each compartment to calculate three different scales of C age and transit time. Therefore, the arithmetic average of C age or transit time at each grid cell (e.g., Fig. 1a) does not equal to the
190 global average of C transit time (e.g. Fig. 3a).

[revised manuscript text omitted]

**Figure 5 a) Changes of global C turnover time (stock-over-flux) in three scenarios, S1: climate warming scenario (red line); S2: rising atmospheric [CO₂] scenario (green line), and S3: Combination of climate warming and rising atmospheric [CO₂] scenario (blue line). b) The deviation of the change in C turnover time ($\Delta\tau_{oto}$) is estimated relative to the change in C transit time ($\Delta\tau_{Rts}$): ($|\Delta\tau_{oto}| - |\Delta\tau_{Rts}|$). Positive indicates more change in C turnover time than C transit time. Grey line represents the reference of no deviation. c) The relative deviation of the change in C turnover time in year 2000 and 2100 is also estimated relative to the change in C transit time: $\frac{(|\Delta\tau_{oto}| - |\Delta\tau_{Rts}|)}{|\Delta\tau_{Rts}|} \times 100\%$.**

595

[Figure]

**Figure 6** a) Latitudinal variation in C transit time ($\tau_{Rts}$) at steady state and b)-c) its change are compared to d)-f) C turnover time ($\tau_{oto}$). The changes between 2090s and 1900s are estimated by c), f) absolute value: $\Delta\tau = (\tau_{2090s} - \tau_{1900s})$ and by b), e) relative value: $\Delta\tau_r = \frac{\Delta\tau}{\tau_{1900s}}$. g) The deviation of C turnover time in relative to C transit time is estimated by $(\tau_{oto} - \tau_{Rts})/\tau_{ts}$ at steady state. In relative to C transit time, the deviation of the change in C turnover time are estimated by h) absolute deviation ($|\Delta\tau_{oto}| - |\Delta\tau_{Rts}|$) and i) relative deviation in $\frac{(|\Delta\tau_{oto}| - |\Delta\tau_{Rts}|)}{|\Delta\tau_{Rts}|}$. All variables are compared in three scenarios: S1: only climate warming scenario (red line); S2: rising atmospheric [$CO_2$] scenario (green line), and S3: Combination of climate warming and rising atmospheric [$CO_2$] scenario (blue line). Grey lines in b), c), e) and f) represent the reference lines of no change and those in h) and i) represent reference line of no deviation.